# Between synchrony and turbulence: intricate hierarchies of coexistence patterns

Sindre W. Haugland [1], Anton Tosolini[1] & Katharina Krischer [1]✉

Coupled oscillators, even identical ones, display a wide range of behaviours, among them synchrony and incoherence. The 2002 discovery of so-called chimera states, states of coexisting synchronized and unsynchronized oscillators, provided a possible link between the two and definitely showed that different parts of the same ensemble can sustain qualitatively different forms of motion. Here, we demonstrate that globally coupled identical oscillators can express a range of coexistence patterns more comprehensive than chimeras. A hierarchy of such states evolves from the fully synchronized solution in a series of cluster-splittings. At the far end of this hierarchy, the states further collide with their own mirror-images in phase space – rendering the motion chaotic, destroying some of the clusters and thereby producing even more intricate coexistence patterns. A sequence of such attractor collisions can ultimately lead to full incoherence of only single asynchronous oscillators. Chimera states, with one large synchronized cluster and else only single oscillators, are found to be just one step in this transition from low- to high-dimensional dynamics.

[1] Physics Department, Nonequilibrium Chemical Physics, Technical University of Munich, James-Franck-Str. 1, D-85748 Garching, Germany.
✉email: krischer@tum.de

One of the big problems in physics is how high-dimensional disorder in space and time may emerge from a spatially ordered, in the simplest case uniform, state with low-dimensional dynamics[1]. Exploring different paths from order to spatiotemporal disorder and their universal character is central for a deeper understanding of complex emergent behaviour such as spatiotemporal chaos in reaction-diffusion systems[2,3] or turbulence in hydrodynamic flows[4,5].

Ensembles of coupled oscillators are one class of apparently simple dynamical systems that yet may adopt states ranging from full synchrony to complete incoherence, and which has provided insights in virtually any discipline, ranging from the natural sciences to sociology[6,7]. During the last two decades, a kind of hybrid phenomenon, in which synchronized and incoherent oscillators coexist in an ensemble of identical oscillators[8], coined a chimera state[9], has received considerable attention (see reviews[10–12] and the references therein), not least since it can be considered a "natural link between coherence and incoherence"[13]. In an earlier study employing globally coupled logistic maps[14], four different classes of behaviour were found, including a large variety of partially ordered states, some of which were later classified as chimeras[15]. Yet, the bifurcation structure between the different classes was not resolved.

In this article, we study the bifurcations from synchrony, via clustered and partially clustered states to full incoherence in a system of globally coupled oscillators with nonlinear coupling, with simulations and bifurcation analysis for an increasing number of oscillators. Here, chimera states are just one of a multitude of coexistence patterns, all consisting of clusters, that is, internally synchronized groups of oscillators, of widely different sizes and dynamics, and possibly including one or several single oscillators. The path towards complete incoherence begins with a symmetry-breaking cascade of cluster-splitting period-doubling bifurcations, wherein the currently smallest cluster is repeatedly split into two, leading to hierarchical clustering. Due to the high symmetry of the system, each symmetry-breaking produces many equivalent mirror-image variants of each outcome state, multiplying the number of attractors and leading to an ever more crowded phase space[16]. At some point, each variant collides with some of its mirror-images, creating larger attractors with higher symmetry. Usually, this blows up some of the clusters, the resulting single oscillators henceforth moving similarly on average. A succession of such symmetry-increasing bifurcations destroys first the smallest clusters, and then the larger ones, partially mirroring the former cluster-splitting cascade and ultimately creating a completely incoherent state. A chimera state, consisting of one synchronized cluster and otherwise only single, incoherent oscillators is often the second to last state of the sequence.

The model we employ is an ensemble of $N$ Stuart-Landau oscillators $W_k \in \mathbb{C}$, $k = 1, \ldots, N$, with nonlinear global coupling[17]:

$$\frac{dW_k}{dt} = W_k - (1 + ic_2)|W_k|^2 W_k$$
$$- (1 + i\nu)\langle W\rangle + (1 + ic_2)\langle |W|^2 W\rangle, \quad (1)$$

where $c_2$ and $\nu$ are real parameters and $\langle \ldots \rangle = 1/N\sum_{k=1}^{N} \ldots$ denotes ensemble averages. The Stuart-Landau oscillator itself is a generic model for a system close to a Hopf bifurcation, that is, to the onset of self-sustained oscillations[18]. Networks of such oscillators have previously been found to exhibit a wide range of dynamics, many of them occurring for linear global coupling[19–23]. The nonlinear global coupling in Eq. (1) stands out by featuring two qualitatively different chimera states, each of them deduced to somehow emerge from a corresponding type of two-cluster solution[24]. Originally, this coupling was inspired by

electrochemical experiments, wherein the oxide layer on a silicon electrode displays a wide range of spatiotemporal patterns[17]. A few experimental measurements reminiscent of new results in Eq. (1) will be discussed later in this article.

Because the oscillators are identical and the coupling is global, the system is $\mathbb{S}_N$-equivariant: If $\mathbf{W}(t) \in \mathbb{C}^N$ is a solution, then so is $\gamma\mathbf{W}(t) \, \forall \, \gamma \in \mathbb{S}_N$, where $\mathbb{S}_N$ is the symmetric group of all permutations of the $N$ oscillators[25]. Or in less mathematical terms: If we start at a solution to Eq. (1) and interchange the trajectories of any two oscillators, the result is still a solution. Further, the average $\langle W\rangle$ is confined to simple harmonic motion with frequency $\nu$, as shown by taking the ensemble average of the whole equation:

$$\left\langle \frac{dW_k}{dt} \right\rangle = \frac{d}{dt}\langle W\rangle = -i\nu\langle W\rangle \; \Rightarrow \; \langle W\rangle = \eta \, e^{-i\nu t}, \quad (2)$$

where $\eta \in \mathbb{R}$ is an additional parameter, implicitly set by choosing the initial condition. This constraint also implies that for a Poincaré map[26] defined by sampling the system with frequency $\nu$, the average of the $N$ components of the map will always be constant. Thus the nonlinear constraint in the time-continuous Eq. (1) becomes a linear constraint in the time-discrete map.

## Results

The fully synchronized solution $W_k = \eta e^{-i\nu t} \, \forall k$ always exists and is stable for sufficiently large values of $\eta$. It loses stability in either an equivariant pitchfork bifurcation, producing separate clusters that continue to orbit the origin with frequency $\nu$ at different fixed amplitudes, or an equivariant Hopf bifurcation to a $T^2$ torus, producing separate modulated-amplitude clusters that henceforth oscillate with two superposed frequencies $\nu$ and $\omega_H$[27]. We will focus on the latter and the dynamics arising from these.

The equivariant Hopf bifurcation occurs at $\eta_H = 1/\sqrt{2}$ for suitable values of $c_2$ and $\nu$. For $\nu = 0.1$, which we keep fixed throughout, it does for $c_2 < -0.448$[17]. In this Hopf bifurcation, differently balanced two-cluster solutions ranging from $(N - 1) - 1$ (with all but one oscillator in the largest cluster) to $N/2 - N/2$ (with half the oscillators in each cluster) emerge from the synchronized solution. Some of these emerge as stable and others as unstable, depending on the value of $c_2$. The balanced $N/2 - N/2$ solution, with an equal number of oscillators in each cluster, is shown in Fig. 1a, b. The dashed circle marks the enforced path of the ensemble average $\langle W\rangle = \eta e^{-i\nu t}$, which the two clusters orbit on opposite sides as it circles the origin. An unbalanced $3N/4 - N/4$ solution, with $N_1 = 3N/4$ of the oscillators in one cluster and $N_2 = N/4$ in the other, looks as in Fig. 1c, d.

Because $\langle W\rangle$ is independent of the individual oscillator dynamics, the value of any oscillator in the frame of reference of the ensemble average is always given by the simple transformation

$$W_k = \eta \, e^{-i\nu t}(1 + w_k) \; \Rightarrow \; w_k = W_k\eta^{-1}e^{i\nu t} - 1, \quad (3)$$

where $w_k$ is the value of $W_k$ in the co-rotating frame. There, the $N/2 - N/2$ solution from Fig. 1a, b is simply periodic with frequency $\omega_H$ and looks as in Fig. 1e, f. An unbalanced modulated-amplitude $3N/4 - N/4$ solution like that in Fig. 1c, d appears as in Fig. 1g, h. The average of all oscillators in the co-rotating frame of $\langle W\rangle$ is of course always zero. Notably, the global coupling ensures that all solutions for an ensemble size $N$ are also solutions for $N' = nN$, $n \in \mathbb{N}$, with every cluster scaled up by a factor of $n$. For solutions that contain only clusters $N_i \geq 2$, the stability properties will also be the same for different $n$[22,28].

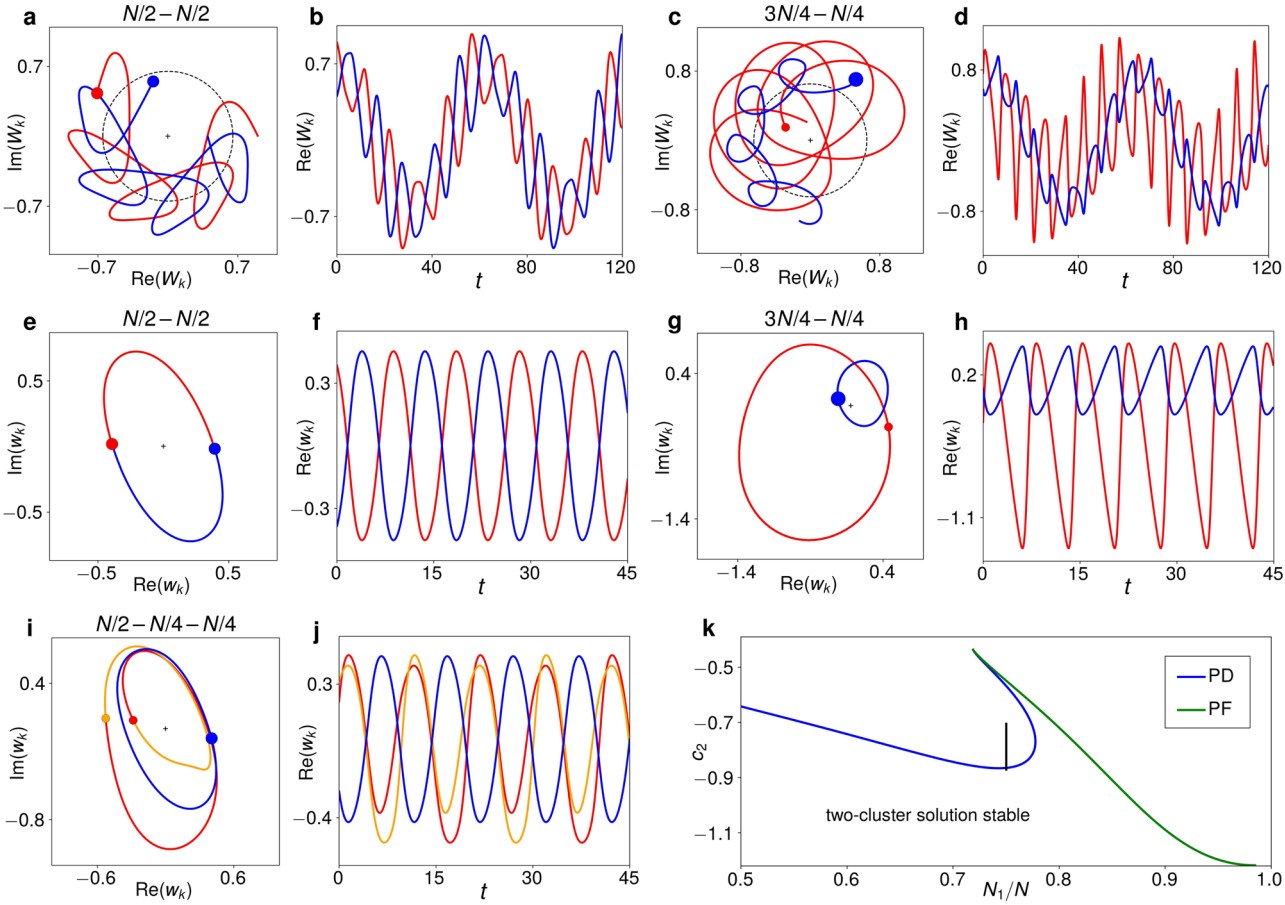

**Fig. 1 Two-cluster solutions and their bifurcations.** Each filled circle refers to the instantaneous position of a cluster, the line of the respective same colour to its trajectory. **a** Trajectory of an $N/2 - N/2$ modulated-amplitude cluster solution for $c_2 = -0.71$ and $\eta = 0.65$. **b** Time series of the real part of each oscillator in **a**. **c, d** Like **a, b** for an unbalanced $3N/4 - N/4$ solution at $c_2 = -1.0$ and $\eta = 0.65$. **e–h** The solutions in **a, b** and **c, d**, respectively, when viewed in a frame co-rotating with the ensemble average $\langle W \rangle = \eta\, e^{-i\nu t}$. For the $N/2 - N/2$ solution, the two clusters follow the same trajectory. **i, j** $N/2 - N/4 - N/4$ three-cluster solution at $c_2 = -0.69$, emerging from the solution in **a, b** in a period-doubling bifurcation. **k** Bifurcations destabilizing the two-cluster solution as a function of $c_2$ and the relative size of the larger cluster $N_1/N$ for $\eta = 0.67$. The blue line denotes a period-doubling (PD) that splits the smaller cluster, and the green line a subcritical pitchfork (PF) that blows it up. The vertical black line marks the $c_2$-incremented simulation in Fig. 6.

If we initialize the $N/2 - N/2$ solution at a point in the $c_2 - \eta$ parameter plane were it is stable and from there on gradually change $c_2$ and/or $\eta$ appropriately, one of the two clusters will break up into two smaller clusters. A possible outcome is shown in Fig. 1i, j. The trajectory of the two new clusters is no longer simply periodic, but period-2, with a small and a large loop. The $N/2 - N/2$ solution has thus become unstable in a symmetry-breaking period-doubling bifurcation, giving rise to a stable $N/2 - N/4 - N/4$ three-cluster solution. This bifurcation also destabilizes less balanced two-cluster solutions, such as the $3N/4 - N/4$ solution in Fig. 1g, h. In these cases, the smaller of the two clusters is split. The position of of the period-doubling bifurcation in parameter space depends on the relative sizes of the clusters, as shown by the blue line in Fig. 1k, which tracks the value of $c_2$ at which this bifurcation occurs as a function of $N_1/N$ for $\eta = 0.67$.

For very unbalanced solutions $N_1/N > 0.8$, the smallest cluster is destroyed in a subcritical pitchfork bifurcation (green line). This results in several smaller clusters and/or single oscillators, depending on the relative sizes of the initial two clusters. In some cases, a few oscillators originally in the smaller cluster are also absorbed by the larger one. As the transition is subcritical, these outcome states are not directly related to the initial two-cluster

solution, but rather belong to a different, coexisting solution branch. They will not concern us further here.

**Hierarchical clustering through pervasive stepwise symmetry breaking.** If we concentrate on the $N/2 - N/2$ solution, that is, keep $N_1/N = 0.5$ fixed, we can track the cluster-splitting period-doubling bifurcation in both $c_2$ and $\eta$ simultaneously. A part of the resultant bifurcation line in the $c_2 - \eta$ parameter plane is delineated by the leftmost line in Fig. 2c. Beyond this bifurcation, we find a mesh of additional cluster-splitting bifurcation curves, creating a hierarchy of successively less symmetric multi-cluster solutions with various periodicities. Each bifurcation involves the breakup of either one cluster or two similarly behaving clusters and produces several qualitatively different solutions, differing by how the oscillators of the splitting cluster(s) distribute. (For example, the $4 - 4$ solution for $N = 8$ can split into either $4 - 2 - 2$, $4 - 3 - 1$, $2 - 2 - 2 - 2$, $2 - 2 - 3 - 1$ or $3 - 1 - 3 - 1$.) However, all these solutions will usually not be co-stable.

Figure 2c shows the stability boundaries of several solutions for $N = 16$. The $N/2 - N/2 = 8 - 8$ solution is stable in the upper left. This solution is destabilized at the leftmost blue period-doubling line. When increasing $c_2$ past this line for $\eta > 0.635$ (that is, in the

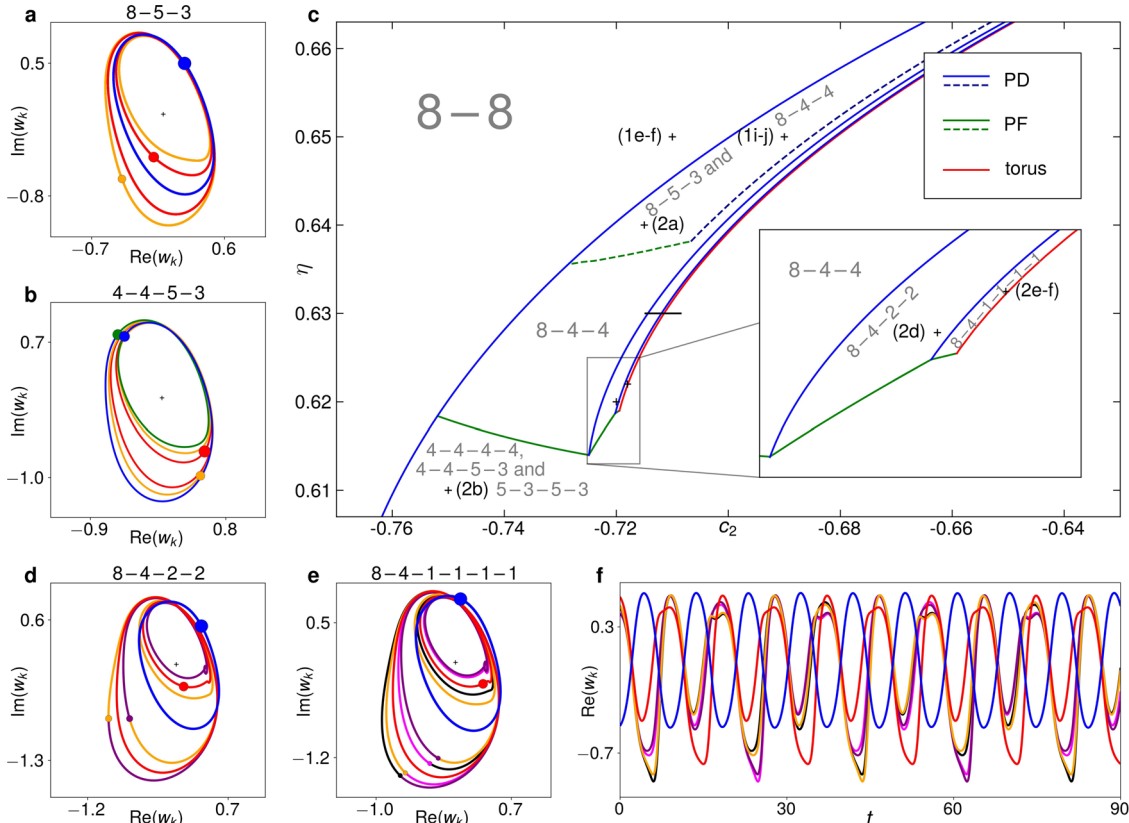

**Fig. 2 Cluster solutions and bifurcations for $N = 16$.** Each filled circle refers to the instantaneous position of a cluster or single oscillator, the line of the respective same colour to its trajectory. Sizes of filled circles mirror sizes of clusters. **a** $8 - 5 - 3$ solution in the co-rotating frame of $\langle W \rangle$ for $c_2 = -0.715$ and $\eta = 0.64$. **b** $4 - 4 - 5 - 3$ solution for $c_2 = -0.75$ and $\eta = 0.61$. **c** Bifurcation diagram of $8 - 8$-derived solutions with PD period-doubling, PF pitchfork. Dashed lines mark bifurcations of the $8 - 5 - 3$ solution (**a**), solid lines those of the $8 - 4 - 4$ (Fig. 1i-j) and the solutions emerging from it. Grey labels mark where each type of solution is stable. Labeled crosses mark the parameter values of the solutions depicted in Fig. 1e, f, i, j, **a**, **b** and **d**–**f**. The black line marks the $c_2$-incremented simulation in Fig. 3a, b. **d** $8 - 4 - 2 - 2$ solution for $c_2 = -0.72$ and $\eta = 0.62$. **e**, **f** $8 - 4 - 1 - 1 - 1 - 1$ solution for $c_2 = -0.718$ and $\eta = 0.62$, including time series of real part of each cluster and single oscillator.

upper half of the figure), it gives rise to stable $8 - 4 - 4$ and $8 - 5 - 3$ solutions (shown in Figs. 1i, j and 2a, respectively). The $8 - 5 - 3$ solution is stable within the two dashed lines. Below the dashed green line, this solution in turn produces a stable $5 - 3 - 5 - 3$ and unstable $5 - 3 - 6 - 2$ and $5 - 3 - 7 - 1$ solutions. At the dashed blue line, it undergoes another period-doubling cluster split to an $8 - 5 - 2 - 1$ period-4 solution.

Between $\eta = 0.62$ and $\eta = 0.635$, only the $8 - 4 - 4$ solution emerges as stable when crossing the leftmost period-doubling line. The remaining solid bifurcation lines all affect this solution and its descendants. At the solid green line from $c_2 \approx -0.755$ to $c_2 \approx -0.725$ in the lower left, it produces stable $4 - 4 - 4 - 4$ and $4 - 4 - 5 - 3$ (Fig. 2b) solutions, as well as unstable $4 - 4 - 6 - 2$ and $4 - 4 - 7 - 1$ solutions. Like the dashed green line, this is an equivariant pitchfork bifurcation, splitting clusters, but not altering the overall periodicity of the ensemble. Below this pitchfork line, the abovementioned $4 - 4 - 4 - 4$, $4 - 4 - 5 - 3$, $4 - 4 - 6 - 2$ and $4 - 4 - 7 - 1$ four-cluster solutions also emerge directly from the $8 - 8$ solution at the leftmost period-doubling line.

At the solid blue line directly to the right of the dashed blue one, the $8 - 4 - 4$ solution undergoes a period-doubling bifurcation analogous to that of the $8 - 5 - 3$ solution, producing a stable $8 - 4 - 2 - 2$ (Fig. 2d) and an unstable $8 - 4 - 3 - 1$ period-4 solution. The former becomes unstable either at the bottom diagonal green pitchfork line at $c_2 \approx -0.72$ or at the rightmost blue period-doubling line. In the latter case (see inset), the $8 - 4 - 2 - 2$ solution

produces an unstable $8 - 4 - 2 - 1 - 1$ and a stable $8 - 4 - 1 - 1 - 1 - 1$ period-8 solution (Fig. 2e, f).

At the red line in Fig. 2c, the $8 - 4 - 1 - 1 - 1 - 1$ solution undergoes a torus bifurcation, whereby a third frequency is added to the dynamics, while all clusters stay intact. The resultant three-frequency motion is resistant to the addition of small random numbers over a nonzero $c_2$ interval. This is notable as stable quasiperiodic dynamics with more that two frequencies is usually not observed. It has even been proven that quasiperiodic dynamics with three or more frequencies are in general structurally unstable[1,29]. However, such stable quasiperiodic motion on $T^3$ has also been observed in Stuart-Landau oscillators with linear global coupling[20] and could be due to the rotational invariance of the differential equations.

If we initialize the $8 - 4 - 4$ solution at $c_2 = -0.71$ and $\eta = 0.63$ and slowly increase $c_2$ along the horizontal black line in Fig. 2c, the maxima of $\mathrm{Re}(w_k)$ for $k = 1, \ldots, 16$ develop as in Fig. 3a: Initially, there are one maximum of the oscillators in the cluster of eight (blue) and two shared maxima of the two period-2 clusters of four (red). When one of these clusters splits up into two smaller clusters of two at the period-doubling bifurcation $PD_2$, the maxima of these smaller clusters henceforth appear as four distinct yellow lines. In the next period-doubling bifurcation ($PD_3$), these lines split up into eight.

From the fully synchronized solution to the $8 - 4 - (4 \times 1)$ solution, four discrete steps of symmetry breaking have taken place: one initial equivariant Hopf bifurcation, as well as three equivariant

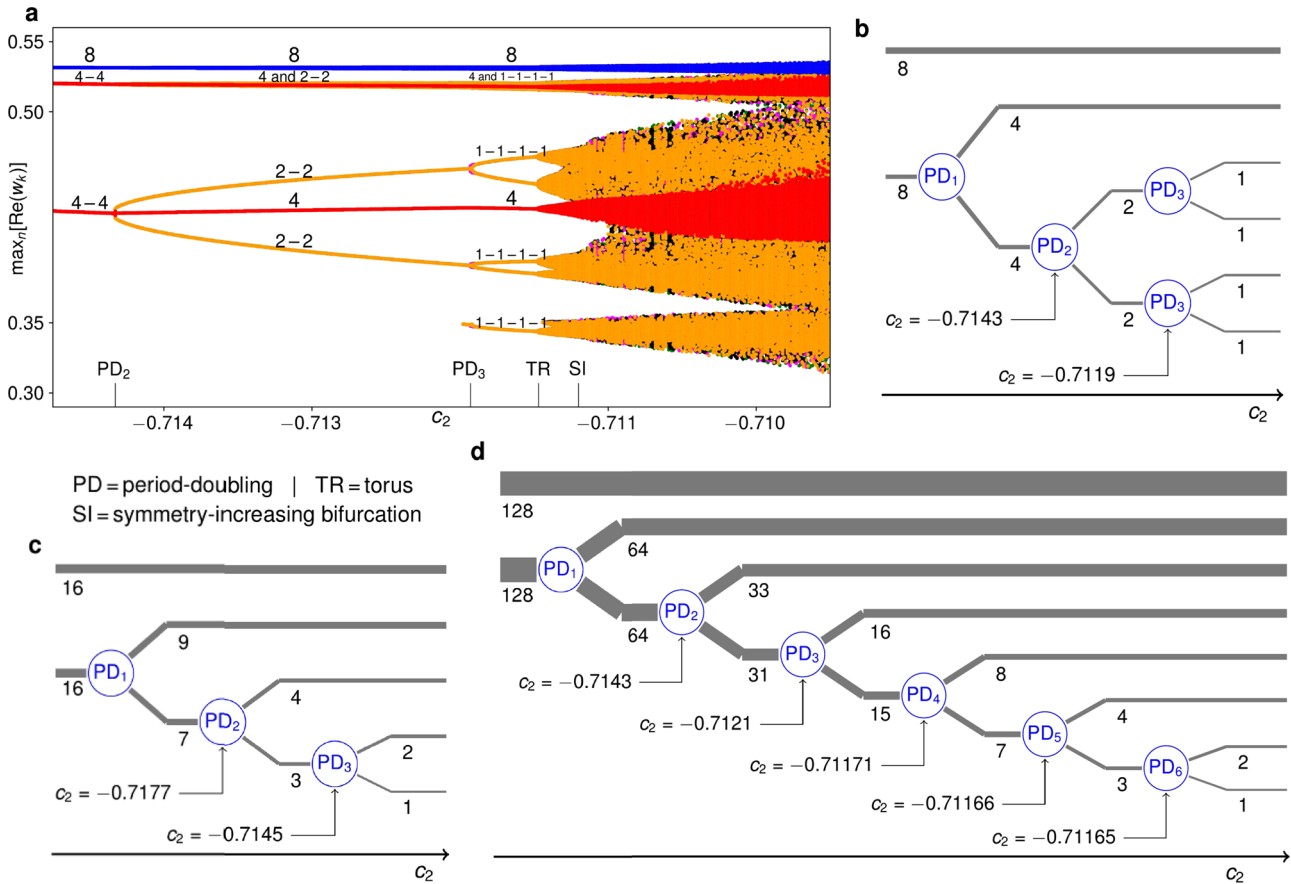

**Fig. 3 Cluster-splitting cascades for different N. a** Maxima of $Re(w_k)$ in an $N = 16$ $c_2$-incremented simulation at $\eta = 0.63$. Labels on the figure mark the clusters reaching the different maxima as the solution changes from $8 - 4 - 4$ via $8 - 4 - 2 - 2$ to $8 - 4 - (4 \times 1)$. Labels on the abscissa mark occurring bifurcations. The additional smallest yellow maximum appearing at $c_2 \approx -0.712$ is caused by the continuous deformation of the oscillator trajectories and not by a bifurcation. At $c_2 \approx -0.7115$, the torus bifurcation to three-frequency dynamics manifests itself in a distinct broadening of the formerly discrete maxima. **b** Schematic portrayal of the period-doubling cluster splittings in the simulation in **a**. **c, d** Like **b** for $c_2$-incremented simulations at $\eta = 0.63$ with $N = 32$ and $N = 256$, respectively, based on quantitative results shown in Supplementary Figs. 1–3 and described in Supplementary Note 1.

period-doubling bifurcations. The three last of these steps are shown schematically in Fig. 3b. Similar stepwise symmetry breaking is observed both for larger $N$ and when the smallest cluster does not break up into equal-sized parts (Fig. 3c). The larger $N$ is, the more steps occur, at ever closer parameter values, and for $N = 256$, as many as seven steps can be observed (see Fig. 3d). The $N/2 - N/2$ two-cluster solution thus gives rise to a cluster-splitting cascade, producing a multitude of coexisting multi-cluster states and, most notably, hierarchical clustering.

**Symmetry-increasing bifurcation and temporary clusters**. At the end of a cascade of cluster-splitting period-doubling bifurcations, a torus bifurcation usually occurs (see e.g. the red bifurcation line in Fig. 2c). The resultant $T^3$ motion is usually stable for a nonzero parameter interval, before being superseded by less regular dynamics in a symmetry-increasing bifurcation[30], wherein several distinct, but equivalent variants of the same solution collide. These variants exist because Eq. (1) is $\mathbb{S}_N$-equivariant. Thus, any solution remains a solution when any of the oscillators are interchanged, and each solution (except the fully synchronized one) exists in the form of several distinct symmetric variants in phase space. (For example, if we interchange an oscillator from the blue cluster in Fig. 1a with one from the red, the outcome is such a different, but equivalent variant.)

All solutions investigated here are at least periodic in the co-rotating frame. The attractor corresponding to a stable solution thus occupies more than a single point in the phase space spanned by $w_k, k = 1, ..., N$. As these attractors become more complex, and especially as the aforementioned torus bifurcation renders the motion quasiperiodic, the part of phase space they occupy increases in extent. This of course applies equally to all the symmetrized variants of each solution.

At some point, two or more variants might grow to touch each other in phase space. When this happens, the variants involved in the collision merge to become a single instance of a new solution, of which there are fewer distinct mirror-image variants in total. The attractor on which the new solution lives is correspondingly more symmetric than the attractors of the colliding variants. One symmetry-increasing bifurcation can in general be followed by another, further increasing the attractor symmetry.

In the $N = 16$ case in Fig. 3a, the first symmetry-increasing bifurcation only disrupts the former rigid cyclic order of the four single oscillators, inherited from the solution in Fig. 2e, f (i.e. that the purple oscillator trails the yellow one, which trails the pink, and so on). In other cases, some of the intact clusters of a certain colliding variant contain oscillators that are in a different cluster in some of the other variants this variant is colliding with. Then, the symmetry-increasing bifurcation destroys these clusters. Such a scenario is schematically shown in Fig. 4: In this $N = 8$ example

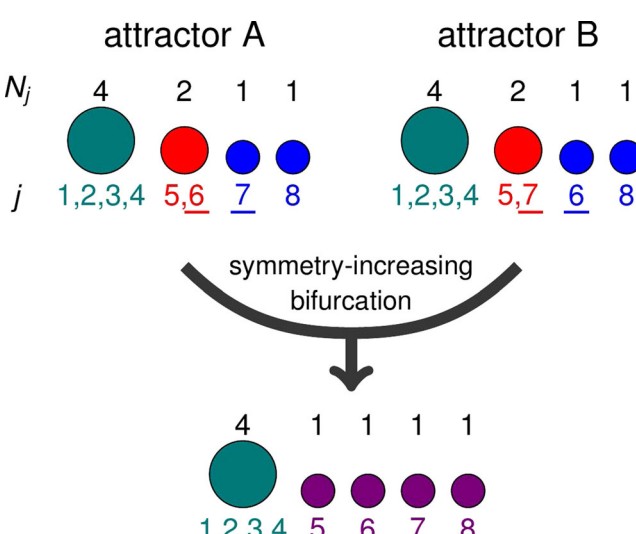

**Fig. 4 Schematic of a symmetry-increasing bifurcation destroying clusters.** Here, two equivalent variants of a $4 - 2 - 1 - 1$ solution collide. The increase in attractor symmetry caused by the collision implies that any two oscillators which behaved equivalently in either of the variants must also behave equivalently in the resulting solution. Because the cluster of two contains different oscillators in the two variants, this cluster is necessarily destroyed. $N_j$ denotes cluster sizes and $j$ denotes the index of each oscillator.

of two colliding $4 - 2 - 1 - 1$ variants, the cluster of two in one of the variants contains oscillators 5 and 6, while in the other, it contains oscillators 5 and 7. Because the two variants are identical mirror-images of each other, they must both be treated equally by the collision. Thus, the oscillators 5 and 6, which are clustered in only one of the variants, cannot remain together after the collision, nor can the oscillators 5 and 7. The result is thus a $4 - 1 - 1 - 1 - 1$ state in which all four single oscillators behave identically. Similarly for larger ensembles, as the attractor symmetry is increased, the number of single oscillators, in general, grows, in a sense also decreasing the overall order of the ensemble.

The $N = 32$ cluster-splitting cascade in Fig. 3c is also followed by symmetry-increasing bifurcations, and at some point, the long-term cluster-size distribution becomes $16 - 9 - (7 \times 1)$. A time series of the resulting solution is shown in Fig. 5a: Here, the seven single oscillators in yellow move similarly to the clusters of four, two and one in the former $16 - 9 - 4 - 2 - 1$ solution, being close to deep minima when the red cluster of nine is at a shallow minimum and vice versa. They also repeatedly congregate into loose temporary agglomerations of four, three and two oscillators, respectively. This is further illustrated by Fig. 5b–d, where the cross-correlation between all oscillator trajectories is calculated every $10^4$ time steps. Two oscillators are said to be in the same cluster if their cross-correlation is greater than $1 - \varepsilon$ for $\varepsilon = 10^{-8}$ (b), $\varepsilon = 10^{-4}$ (c) or $\varepsilon = 10^{-2}$ (d). Sometimes, a temporary cluster of three detected for a certain $\varepsilon$ becomes a cluster of four for larger values of $\varepsilon$, such as the blue cluster at $t = 7 \cdot 10^4$ and the red cluster at $t = 1.2 \cdot 10^5$. This means that four oscillators are loosely congregating here, but that three of the oscillators are more strongly clustering than the fourth. The ensemble is thus less closely approaching the remains of a formerly stable $16 - 9 - 4 - 2 - 1$ attractor in phase space.

Dynamics like those in Fig. 5 have previously been observed by Kaneko in globally coupled logistic maps when the phase space becomes so full of mirror-image attractors that they inevitably intrude upon each other[31]. The outcome is a form of chaotic itinerancy[32], wherein the system meanders between the attractor

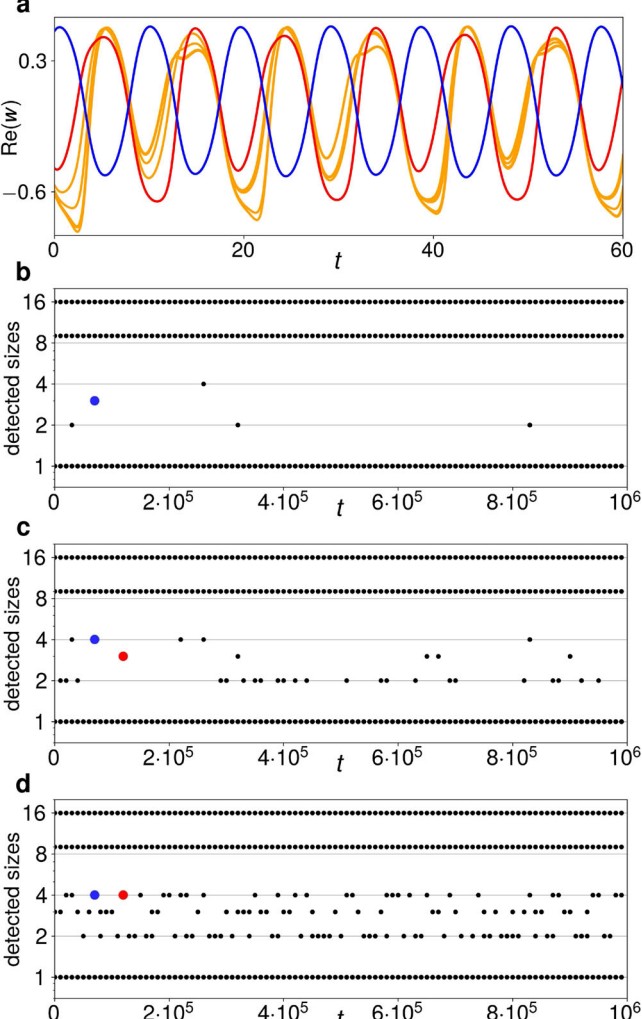

**Fig. 5 Partially clustered $16 - 9 - (7 \times 1)$ solution past the cascade in Fig. 3c. a** Time series of $\mathrm{Re}(w_k)$ for $c_2 = -0.712$ and $\eta = 0.63$. The single oscillators (yellow) reach a deep minimum approximately when the cluster of nine (red) reaches a shallow minimum and the cluster of 16 (blue) reaches a maximum. **b–d** Cluster sizes detected when two oscillators are said to be clustered if their cross-correlation over an interval of 800 time units is $> 1 - \varepsilon$ for $\varepsilon = 10^{-8}$, $10^{-4}$ and $10^{-2}$, respectively. The large red and blue dots mark two exemplary loose oscillator conglomerations at different sensitivities $\varepsilon$.

ruins of previous attractors, each of them relatively low-dimensional, but connected by higher-dimensional transitional motion[15].

Also found in globally coupled maps is precision-dependent clustering, wherein trajectories of individual maps that are unclustered when distinguished with high precision appear to repeatedly merge into the ever thicker branches of a clustering tree when the precision is decreased[14]. In our ensemble, this occurs as a consequence of the symmetry-breaking period-doubling cascade. For example, past the $N = 256$ cascade in Fig. 3 (at $c_2 \approx -0.71162$), we encounter a $128 - 64 - 33 - (31 \times 1)$ itinerant solution that for small $\varepsilon \le 10^{-5}$ is found to have an additional cluster of usually 16, sometimes 18 or 19 oscillators, while the remaining oscillators repeatedly form ephemeral smaller clusters of strongly fluctuating sizes. For $\varepsilon = 10^{-4}$, a cluster of size 15 is also sometimes detected (along with that of 16), and for $\varepsilon = 10^{-3}$ the sizes are always $128 - 64 - 33 - 16 - 15$, $128 - 64 - 33 - 18 - 13$ or $128 - 64$

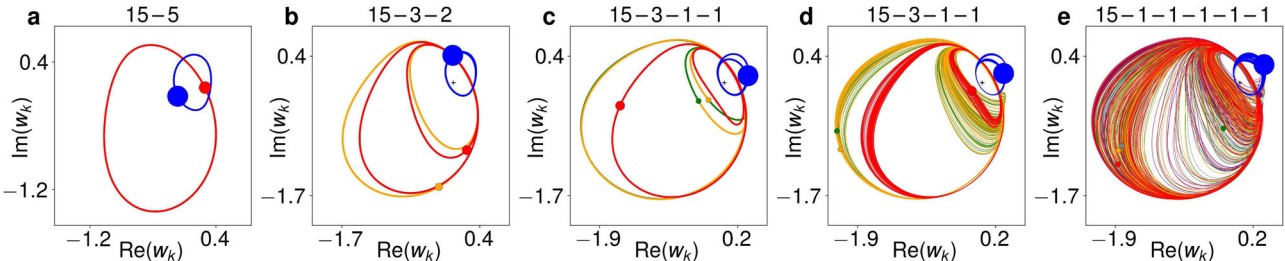

**Fig. 6 Steps from $15-5$ two-cluster solution to chimera state.** Each filled circle refers to the instantaneous position of a cluster or single oscillator, the line of the respective same colour to its trajectory. Sizes of filled circles mirror sizes of clusters. $N = 20$, $\nu = 0.1$ and $\eta = 0.67$, proceeding along the black line in Fig. 1k. **a** $15-5$ solution for $c_2 = -0.87$. **b** $15-3-2$ period-2 solution for $c_2 = -0.82$. **c** $15-3-1-1$ period-4 solution for $c_2 = -0.725$. The two largest loops of the two single oscillators are almost equal. **d** $15-3-1-1$ three-frequency solution for $c_2 = -0.71$. **e** $15-(5\times1)$ chimera at $c_2 = -0.705$, after collision of mirror-image attractors.

$-33-19-12$. For $\varepsilon = 10^{-2}$, they are $128-64-33-31$ throughout, and for $\varepsilon = 10^{-1}$, $128-64-64$. The same pattern to some extent already applies in the quasiperiodic domain of Fig. 3b–d, where clusters are most strongly correlated with those other clusters from which they most recently split.

If we initialize the ensemble in the itinerant state beyond a symmetry-increasing bifurcation and gradually change the parameters back towards more regular motion, the transition to the relevant two-cluster solution will simply be the reverse of the one that created the itinerant state. For example, if we initialized the $N = 16$ ensemble in the state at the right edge of Fig. 3a and slowly decreased $c_2$, this would produce the same sequence of bifurcations. See Supplementary Fig. 4 and Supplementary Note 2.

When the equation parameters are incremented too far into the regime of chaotic itinerancy, the ensemble will often jump to an entirely different solution. Beyond the $16-9-7$-derived state in Fig. 5, it e.g. jumps to the blue hitherto co-stable $16-8-8$-derived branch. However, the end result can also be the destruction of all permanent clusters and the motion of only single oscillators on a fully symmetric chaotic attractor. See Supplementary Figs. 5–7 and Supplementary Note 3.

**Emergence of a chimera state.** In our context, a chimera state is an $N_1 - ((N - N_1) \times 1))$ solution. The modulated-amplitude chimeras previously found in Eq. (1) have significantly more synchronized ($N_1$) than unsynchronized ($N - N_1$) oscillators[33]. This suggests they have not evolved from balanced two-cluster solutions like the ones studied above. Yet, our above results can be used to explain how they are created. If we e.g. initialize an $N = 20$ ensemble as an $3N/4 - N/4 = 15 - 5$ solution (Fig. 6a) for $c_2 = -0.87$ and $\eta = 0.67$, the bifurcation diagram in Fig. 1k tells us it will undergo a cluster-splitting period-doubling bifurcation if $c_2$ is increased. The resulting $15 - 3 - 2$ period-2 solution is shown in Fig. 6b. Further up in $c_2$, the cluster of two is split into single oscillators (Fig. 6c). Then, a torus bifurcation smears the previously closed trajectories into continuous bands (Fig. 6d).

Finally, the current $15 - 3 - 1 - 1$ variant collides with nine others in a symmetry-increasing bifurcation. This destroys the cluster of three, resulting in a $15 - (5 \times 1)$ chimera state (Fig. 6e). Note how the three oscillators that are temporarily close to each other in Fig. 6e (red, yellow, grey, in the lower left) are not all the same three that were clustered in Fig. 6b–d (red, purple, grey). The ensemble is currently close to the ruin of a different $15 - 3 - 1 - 1$ solution variant, and the chimera state is thus also an example of chaotic itinerancy. For $N = 200$, the transition from a $150 - 50$ to a $150 - (50 \times 1)$ solution proceeds along a much more involved, but essentially similar path. See Supplementary Fig. 8 and Supplementary Note 4.

**Generality of results I—pitchfork maps**. Other theoretical $\mathbb{S}_N$-symmetric systems can also develop as discussed in the previous sections. One such system is the following ensemble of $N$ globally coupled time-discrete maps:

$$y_k(n+1) = (1 + \alpha - |y_k(n)|^2) \cdot y_k(n)$$
$$- \frac{1}{N}\sum_{j=1}^{N}(\alpha - |y_j(n)|^2) \cdot y_j(n), \quad (4)$$

where $y_k(n) \in \mathbb{R}$ denotes the $n$th iteration of the $k$th map, $k = 1, \ldots, N$, and $\alpha$ is a real-valued parameter. Each map $y_k(n+1) = (1 + \alpha - |y_k(n)|^2) \cdot y_k(n)$ (without the coupling) is modeled on the normal form of the supercritical pitchfork bifurcation [26], $x_{n+1} = x_n + \mu x_n - x_n^3$. Altogether, the system (4) is subject to a conservation law:

$$\langle y(n+1) \rangle = \frac{1}{N}\sum_{k=1}^{N} y_k(n+1) = \langle y(n) \rangle, \quad (5)$$

that is, the ensemble average $\langle y(n) \rangle = \langle y(0) \rangle \, \forall \, n \in \mathbb{N}$ remains constant independent of the individual map behaviour and thus effectively constitutes an additional parameter $\beta = N^{-1}\sum_k y(0)$, implicitly set by choosing the initial value of each map $y_k(0)$.

For suitable values of $\alpha$ and $\beta$, Eq. (4) has stable period-1 (i.e. constant) two-cluster solutions, among them a balanced $N/2 - N/2$ solution for any even $N$. As an example, for $\alpha = 0.7$ and $\beta = 0.15$ this solution is given by $y_k(n) \approx -0.645$ for $k = 1, \ldots, N/2$ and $y_k(n) \approx 0.945$ for $k = N/2 + 1, \ldots, N$. (Of course, any other $N/2$ of the $N$ maps could also be in the first cluster; that would simply constitute a different equivalent variant of the same solution.) Clearly, the ensemble average remains constantly equal to $\beta = 0.15$.

Let us now again consider the concrete case $N = 16$. If $\alpha$ is slowly increased, the $N/2 - N/2 = 8 - 8$ solution undergoes an equivariant period-doubling bifurcation at $\alpha = 0.708$. Like in the Stuart-Landau ensemble, several three-cluster period-2 solutions emerge, one of which is the $N/2 - N/4 - N/4 = 8 - 4 - 4$ solution in Fig. 7a. For a further increase of $\alpha$, this solution also undergoes a period-doubling bifurcation, wherein both clusters of four are split into a total of four period-4 clusters of two, as seen in Fig. 7b. The cluster of eight (at $y_k(n) \approx -0.7$) still remains period-1, but has been left out of the figure for a better view. This sequence of two-cluster splittings, summarized in Fig. 8a, is strongly reminiscent of the two last steps in Fig. 3b.

For sufficiently large ensemble sizes $N$, when the $N/2 - N/2$ solution is destabilized in its period-doubling bifurcation, several of the resultant three-cluster solutions emerge as co-stable. For $N = 128$ and $\beta = 0.15$, one of the stable solutions is a $64 - 33 - 31$ period-2 solution whose trajectory looks more or less like that of the $N/2 - N/4 - N/4$ solution in Fig. 7a. (The maxima of the period-2 trajectory of the cluster of 33 are only slightly smaller than those of the cluster of 31, and its minima slightly less deep, in order to fulfill

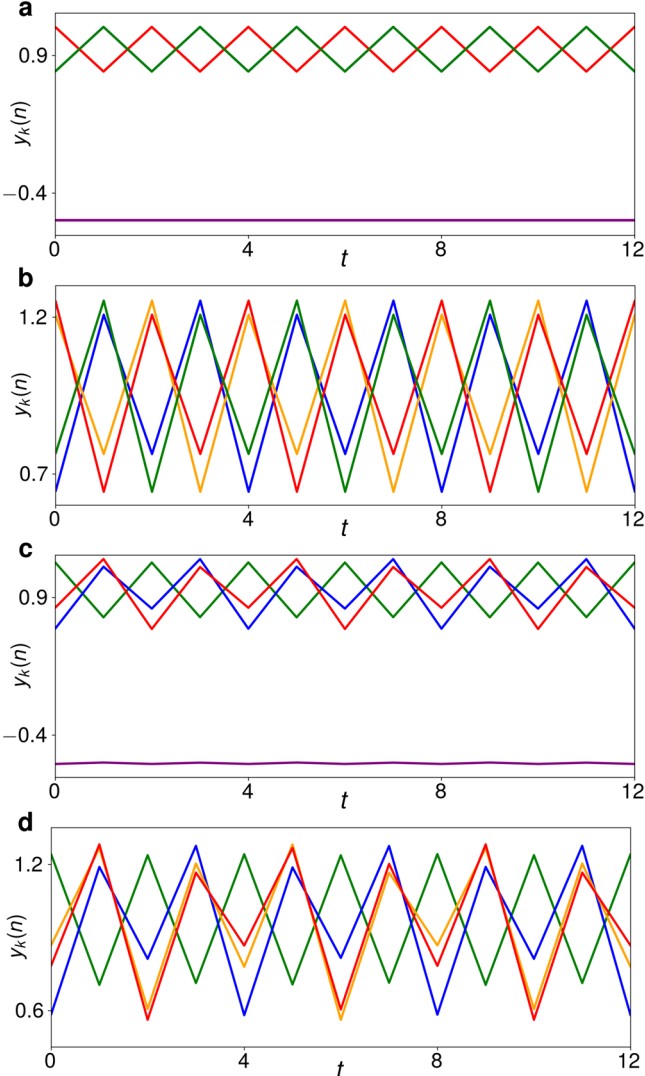

**Fig. 7 Trajectories of time-discrete maps. a** $8 − 4 − 4$ solution for $N = 16$, $\alpha = 0.8$ and $\beta = 0.15$, wherein the clusters of 4 (green and red, respectively) are period-2 and the cluster of 8 (purple) is constant. **b** $8 − (4 \times 2)$ solution emerging in a period-doubling of the solution in **a**, with the cluster of 8 not shown for better resolution. $\alpha = 0.86$. Here, each color marks the trajectory of a cluster of two. **c** $64 − 33 − 16 − 15$ period-4 solution for $N = 128$, $\alpha = 0.86$ and $\beta = 0.15$. Cluster trajectories: 64 purple, 33 green, 16 blue, 15 red. **d** $64 − 33 − 16 − 8 − 7$ period-8 solution emerging in a period-doubling of the solution in **c**, with the cluster of 64 not shown. $\alpha = 0.87$. The cluster of 15 has split into smaller clusters of 8 (red) and 7 (yellow).

the condition $\langle y(n) \rangle = \beta \, \forall n$.) When $\alpha$ is gradually increased for this $64 − 33 − 31$ solution, an equivariant period-doubling also occurs, but here, only the cluster of 31 is split. The result is the $64 − 33 − 16 − 15$ period-4 solution shown in Fig. 7c. (Here, the cluster of 64 also moves with a small period-4 component, due to the asymmetry in the smaller clusters.) If $\alpha$ is increased somewhat further, it results in the $64 − 33 − 16 − 8 − 7$ period-8 solution in Fig. 7d. Even further upward in $\alpha$, two more cluster-splitting period-doubling bifurcations occur, resulting in the overall sequence of cluster sizes shown in Fig. 8b. Thus, Eq. (4) undergoes a cluster-splitting cascade remarkably similar to that of Eq. (1).

The coupled maps of Eq. (4) also exhibit transitions likely to be symmetry-increasing bifurcations. Past the $\alpha$ interval covered in Fig. 8b, the $N = 128$ ensemble namely also enters several

consecutive $\alpha$ intervals wherein the sizes of only some clusters remain stable for several $\alpha$ increments. Other clusters seemingly appear and disappear erratically, as already observed for the Stuart-Landau ensemble in Fig. 5. We even encounter $64 − (64 \times 1)$ chimera states, as shown in Supplementary Fig. 10.

The difference between the two systems (1) and (4) lies in the details. For example, we have already seen that in the Stuart-Landau ensemble, the period-doubling bifurcation of the $N/2 − N/4 − N/4$ solution produces a stable $N/2 − N/4 − N/8 − N/8$ solution, whereas the analogous bifurcation in the coupled maps gives rise to a stable $N/2 − (4 \times N/8)$ solution (comparing Figs. 3b and 8a). Another difference can be observed if we track the $128 − 65 − 63$ period-2 solution to Eq. (4) for $N = 256$ upward in $\alpha$ for $\beta = 0.15$. At first, it will give rise to a stable $128 − 65 − 32 − 31$ period-4 solution. However, the next bifurcation encountered will not be another period-doubling, but an equivariant pitchfork splitting the cluster of 65. Thus, the pattern of stepwise cluster splitting, whereby always the smallest cluster is the next one to be split, as observed in both Figs. 3c, d and 8b, ends prematurely. In this case, no more discrete cluster splittings occur and the next qualitative change of the dynamics is a symmetry-increasing bifurcation, as seen in Supplementary Fig. 11 and described in Supplementary Note 6.

**Generality of results II—electrochemical experiments.** As stated in the introduction, Eq. (1) is inspired by electrochemical experiments. In fact, the theoretical model was originally more complicated, consisting not of discrete identical oscillators, but of a continuous oscillatory medium coupled via both global and local (diffusional) coupling[17,34]. Later results showed that most of the qualitative dynamics obtained in this extended model could still be reproduced if the diffusion was set to zero[33], thus paving the way for our purely globally coupled ensemble. Meanwhile, the experimental system itself has been found to exhibit a vast amount of dynamical phenomena[17,34–41]. Below, we revisit four different spatiotemporal states representative of solutions in the transition scenario outlined above.

The central component of the experiment is an n-type silicon (Si) electrode, immersed in a fluoride-containing electrolyte. A voltage is applied across the electrode, which is also illuminated with a laser. Thus, an oxide layer is grown photo-electrochemically on the Si surface. Simultaneously, the fluoride species in the electrolyte etches away the silicon oxide in a purely chemical process[42]. An ellipsometric setup is used to measure the spatiotemporal changes in the optical pathway through the Si|SiO₂|electrolyte interface[34,35,43].

For a wide range of experimental parameters, the ellipsometric signal can be made to oscillate homogeneously with a simple period[37]. If the parameters are suitably changed, the electrode undergoes a period-doubling bifurcation, resulting in two anti-phase clusters connected by a mediating region with rather low amplitude. An exemplary snapshot of the electrode in this state is shown in Fig. 9a, together with the time series of a section. The location of the section is indicated by the blue line on the image of the electrode. In the depicted snapshot, a rather high ellipsometric signal, displayed by the red color in the upper part of the electrode, coexists with a rather low signal in the lower right, displayed by the blue color. In the time series below, we recognize the oscillation of the ellipsometric signal; the two regions, connected by the cut, oscillate with the same frequency, but in anti-phase to each other. Note that the global time series exhibits a simple periodic oscillation, which, as demonstrated in Supplementary Fig. 12 and described in Supplementary Note 7, defines a rotating frame. Thus, as in the above simulation results, the experimental results are depicted in a rotating frame, that is, the spatial mean of each frame has been subtracted from every point in the same frame.

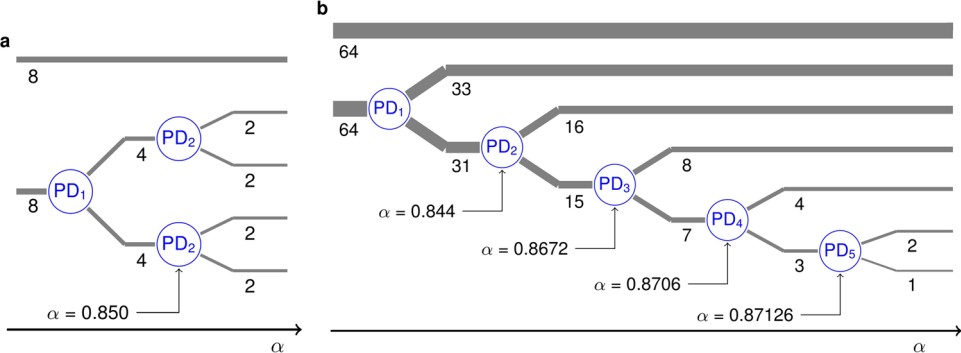

**Fig. 8 Cluster splitting in the time-discrete maps. a** Schematic portrayal of the period-doubling cluster splittings as $\alpha$ is increased for $\beta = 0.15$ and an initial $8 - 8$ solution. **b** Like **a** for a possible cluster-splitting cascade for $N = 128$ and $\beta = 0.15$. The schematics are based on the quantitative results shown in Supplementary Figs. 9 and 10, respectively, and described in Supplementary Note 5.

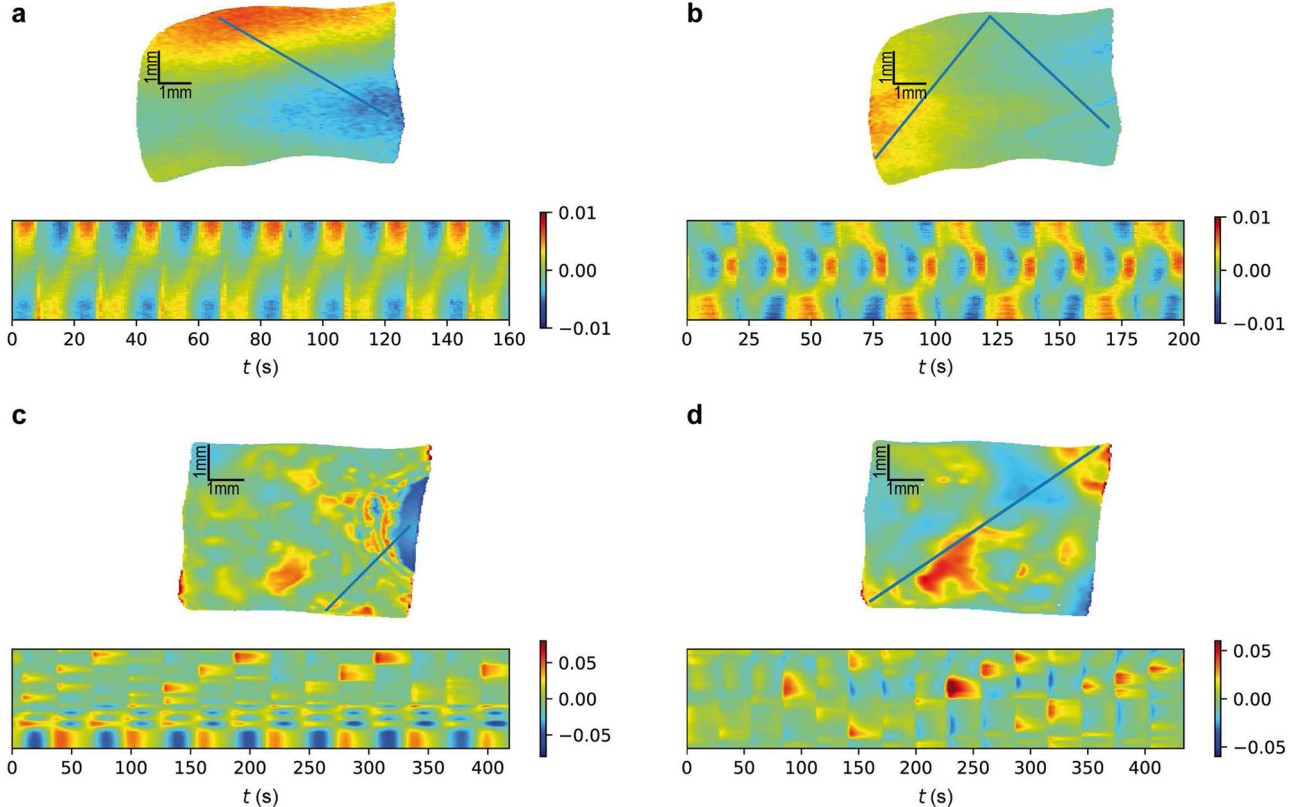

**Fig. 9 Ellipsometric measurements of four different experimental states.** Each state shows a snapshot of the electrode and the time series along a section on the electrode as indicated by the blue lines in the electrode snapshots. Note that the dynamics are shown in a rotating frame, where the uniform oscillation of the spatial average has been subtracted. **a** anti-phase two-cluster state, **b** subharmonic cluster state, **c** chimera state and **d** turbulent state. The size of the electrode in **a** and **b** is 6.0 × 4.3 mm (width × height), while in **c** and **d** it is 5.8 × 4.6 mm. For parameter values and experimental details, see the Methods section.

Figure 9b shows the same electrode after a further parameter variation (see appendix). This time, in order to properly view the spatiotemporal development, the spatial coordinate of the time series is composed of two lines forming an angle. Clearly, the variation of the parameters has resulted in a period-doubling bifurcation affecting the right and left side of the electrode, corresponding to the upper and lower part of the time-series spatial coordinate. These regions now oscillate with double the period of the oscillations in the upper part of the electrode and are in anti-phase with respect to each other.

In Fig. 9c, a deep blue region can be seen on the right of the electrode snapshot. This region appears rather regular throughout whereas the rest of the electrode is irregularly patterned. In the time series of the spatial cut, the deep blue area appears in the lower quarter. It is indeed found to exhibit simply periodic oscillations, whereas most of the electrode is turbulent. This solution is a chimera state.

Finally, Fig. 9d depicts a state that is turbulent throughout. Here, irregular patterns arise over the entire electrode. The time series shows that the spatial incoherence is accompanied by aperiodic behaviour.

Note that the measurements in Fig. 9a, b were carried out on a different day and with a different electrolyte than the ones in Fig. 9c, d. The electrolyte composition seems to be a crucial

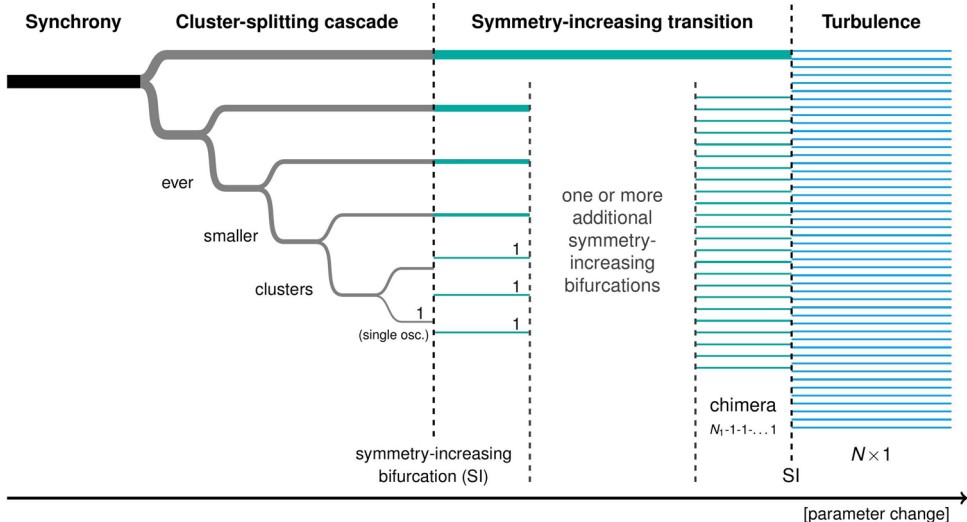

**Fig. 10 Ideal transition from synchrony to turbulence.** The synchronized state is initially broken into two clusters. These clusters are in turn split into successively smaller clusters in a cluster-splitting cascade. When the smallest cluster is just a single oscillator, a symmetry-increasing bifurcation occurs. Hereby, one or more clusters are abruptly broken into single oscillators that henceforth all behave equally in the long run. Additional symmetry-increasing bifurcations ultimately lead to a turbulent state of only single oscillators. The last step but one is often a chimera state.

parameter for some of the presented states, yet it is a parameter which cannot be easily varied during a measurement day. Spatially homogeneous oscillations can, however, be found with both of the electrolytes used here.

## Discussion

In this article, we have shown how a globally coupled system can transition from full symmetry to ever more complex coexistence patterns through a sequence of discrete symmetry-breaking steps. The transition begins with a cascade of cluster-splitting bifurcations, and at each step of this cascade, either one cluster or two similarly behaving clusters are split into smaller clusters. In an ideal form of the cascade, the next cluster to split is always the smallest one, ultimately creating a multi-cluster state with very different cluster sizes, wherein the smallest "cluster" is just a single oscillator. This ideal cascade is schematically depicted in the left part of Fig. 10.

The cluster-splitting cascacde is followed by one or more symmetry-increasing bifurcations, breaking ever more clusters up into single oscillators. Even though they destroy clusters, these bifurcations are symmetry-increasing because the single oscillators they produce all behave equally for $t \to \infty$. Thus, the attractor is symmetric with regards to the interchange of any two of these oscillators.

In practice, the symmetry-increasing transition is often cut short by interactions with other solution branches and the ensemble at some point is thrown onto a different, hitherto co-stable solution. In the ideal case when it is allowed to continue sufficiently long, the end result is ultimately a turbulent state of only single oscillators, all behaving equally in the long run. A chimera state is then the last step but one in the cascade.

Our primary model has been one of $N = 2^n$, $n \in \mathbb{N}$ Stuart-Landau oscillators with nonlinear global coupling and our main focus on the case where the fully synchronized solution is initially split into two equal clusters $N/2 - N/2$. However, the cluster-splitting cascade is also observed for less balanced initial states, such as the $15 - 5$ solution which transitions via $15 - 3 - 2$ to $15 - 3 - 1 - 1$ in Fig. 6. The cascade is also not dependent on the choice of $N = 2^n$, but similarly occurs for odd $N$ as well, as shown in Supplementary Fig. 13 and described in Supplementary Note 8.

Nor is it conditional upon the particular chosen model, but can be similarly observed in globally coupled time-discrete maps.

Symmetry-increasing bifurcations also occur whether the cluster-size distribution of the initial two-cluster state emerging from synchrony is $N_1/N_2 = 1$ (Fig. 3a) or $N_1/N_2 = 3$ (Fig. 6). Moreover, it is observed in both the Stuart-Landau oscillators (1) and the pitchfork maps (4). The general outcome of a symmetry-increasing bifurcation, chaotic itinerancy, has previously been found in globally coupled logistic maps, along with both multi-cluster states, chimeras and precision-dependent clustering, but without an overall explanation of how these phenomena might be bifurcation-theoretically related to each other[14,15,31,32]. This suggests that the bifurcation scenario uncovered here occurs in those logistic maps as well.

What are the prerequisites for the observed bifurcation scenario? The high permutation symmetry of globally coupled equations is probably a central factor shared by the Stuart-Landau oscillators and the pitchfork maps (and Kaneko's logistic maps). $\mathbb{S}_N$ not only has many subgroups, but most of these subgroups have many subgroups as well, and so on. This intricate subgroup structure is mirrored in the hierarchy of successively less symmetric quasiperiodic solutions. Moreover, because $\mathbb{S}_N$ is much larger than those of its subgroups corresponding to the more intricate solutions, there are many mirror-image variants of the latter, causing the symmetry-increasing bifurcations and the itinerant coexistence patterns that these produce.

The nonlinear nature of the global coupling could be another relevant system property, and all three systems studied here are coupled nonlinearly (as are Kaneko's logistic maps). However, symmetry-increasing bifurcations have also been observed in Stuart-Landau oscillators with linear global coupling[44], for an ensemble size as small as $N = 4$. Such an ensemble is of course too small to exhibit an evident cluster-halving cascade, and instead, the symmetry-increasing bifurcation was preceded by a non-equivariant Feigenbaum period-doubling cascade to chaos. To test whether our full bifurcation scenario can occur for linear global coupling as well, is thus an exciting task for future studies.

In the case of the Stuart-Landau oscillators, an additional factor required by the cluster-splitting cascade is the amplitude variation of the cluster orbits. If the clusters were to have fixed amplitudes, i.e., $d|w_k|/dt = 0$ for all oscillators $w_k$, then there could namely

only be three different clusters, due to the Stuart-Landau oscillator being a third-order polynomial[45]. This would render any extended cluster-splitting impossible. On the other hand, the fact that cluster-splitting takes place by means of period-doubling seems to imply that the amplitudes must vary over the course of a full oscillation period.

What all discussed symmetry-increasing bifurcations seem to require, is a certain dimensionality of the dynamics before the bifurcation. (That is, trajectories can for example not be zero-dimensional fixed points or one-dimensional periodic orbits.) In the Stuart-Landau ensemble with nonlinear global coupling and in the pitchfork maps, this extra dimensionality arises in the torus bifurcation at the end of the cluster-splitting cascade. In the aforementioned Stuart-Landau oscillators with linear global coupling, it is provided by a period-doubling cascade to chaos, and in the globally coupled logistic maps, the maps are also in the chaotic regime.

Finally, we again consider the experiments. These exhibit a series of patterns which are similar to the ones in the simulations. The anti-phase clusters in Fig. 9a emerge from the homogeneous oscillation and bring about a second frequency. Figure 9b, c exhibits coexistence patterns, consisting either of clusters of different frequencies (b) or of the coexistence of a regularly oscillating region and irregular motion (c). Figure 9d is a completely turbulent state. These states are clearly reminiscent of the ones found in the simulations, and we are not aware of other bifurcation scenarios that include these states.

Nevertheless, the connection between the experiments and the discussed bifurcation scenario has yet to be clearly demonstrated. As mentioned above, one difficulty is that the electrolyte composition, which is difficult to change within the same experiment, seems to be a crucial bifurcation parameter. Despite the prior equivalence of many results with and without diffusion, the diffusive coupling on the electrode could also potentially influence the dynamical transition. This could for example be investigated in experiments with amorphous instead of crystalline silicon. Altogether, many effects of different parameters on the dynamics are only poorly understood and the detailed oscillation mechanism is still unknown. There is thus great potential for further studies in this direction. The same applies to the search for other experimental systems that exhibit the discussed transition from synchrony to turbulence.

## Methods

**Stuart-Landau oscillators**. The differential equations (1) were solved numerically using the Python programming language[46] (version 2.7 and later 3.8) and the implicit Adams method of the `scipy.integrate.ode` class of the SciPy library[47] (version 1.6) with a time step of $\Delta t = 0.01$. The data were held and processed in the form of NumPy (version 1.19) arrays[48] with complex-valued floating-point elements and visualized using the Matplotlib library and graphics environment (version 3.3) [49]. The numerical results were evaluated using custom-built functions drawing on the resources of these standard Python libraries, written by S.W.H. Simulations were carried out in the non-rotating frame of Eq. (1), and results in the co-rotating frame of $\langle W \rangle$ were visualized applying Eq. (3) to the data after simulations had been carried out. When not otherwise stated, initial conditions of numerical solutions were random numbers on the real line, fulfilling the global constraint that $\langle W \rangle = \eta e^{-ivt}$. This choice was inspired by earlier work[17].

Figures 1e and 2c were created using the dynamical-systems continuation software Auto07p[50,51] to continue solutions in parameter space. As Auto can only continue fixed-point and limit-cycle solutions, Eq. (1) had to be formulated in the co-rotating frame of the ensemble average in order to carry out these continuations, yielding

$$\begin{aligned}
\frac{da_k}{dt} &= a_k - \nu b_k - \eta^2[A_k - c_2 B_k] \\
&\quad + \frac{1}{N}\eta^2 \sum_j [A_j - c_2 B_j], \\
\frac{db_k}{dt} &= b_k + \nu a_k - \eta^2[B_k + c_2 A_k] \\
&\quad + \frac{1}{N}\eta^2 \sum_j [B_j + c_2 A_j],
\end{aligned} \tag{6}$$

where $w_k = a_k + ib_k$ with $a_k, b_k \in \mathbb{R}$, $k = 1, \ldots, N$, and

$$\begin{aligned}
A_k &= 3a_k + 3a_k^2 + a_k^3 + a_k b_k^2 + b_k^2, \\
B_k &= b_k + 2a_k b_k + a_k^2 b_k + b_k^3.
\end{aligned} \tag{7}$$

To obtain Fig. 2c, the relevant $N = 16$ quasiperiodic solutions where first generated using Python simulations. The output data were transferred to the rotating frame, and a time series corresponding to one full period in that frame was used as input for a $c_2$ or $\eta$ one-parameter continuation of each periodic solution, in order to detect the location of the depicted bifurcations. Then, the detected bifurcations were two-parameter continued in $c_2$ and $\eta$ to obtain the depicted bifurcation lines.

To obtain Fig. 1e, Eq. (6) was reduced to a two-cluster model by setting $a_k = a_{c1}$ and $b_k = b_{c1}$ for all $k = 1, \ldots, N_1$, where $w_{c1} = a_{c1} + ib_{c1}$ is the value of the first cluster. All other oscillators $k = N_1 + 1, \ldots, N$ are in the other cluster $w_{c2} = a_{c2} + ib_{c2}$. This yields the following equation for the first cluster

$$\begin{aligned}
\frac{da_{c1}}{dt} &= a_{c1} - \nu b_{c1} \\
&\quad + \eta^2 \left(1 - \frac{N_1}{N}\right)[A_{c2} - A_{c1} - c_2 (B_{c2} - B_{c1})], \\
\frac{db_{c1}}{dt} &= b_{c1} + \nu a_{c1} \\
&\quad + \eta^2 \left(1 - \frac{N_1}{N}\right)[c_2 (A_{c2} - A_{c1}) + B_{c2} - B_{c1}],
\end{aligned} \tag{8}$$

with $A_{c1}$ and $B_{c1}$ analogous to Eq. (7), while $w_{c2} = \frac{N_1}{N-N_1}w_{c1}$, because of the constraint that $\sum_k w_k = 0$. Thus, the reduced two-cluster model is only two-dimensional. The relative size of the first cluster, $N_1/N$, becomes an effective fourth parameter, in addition to $c_2$, $\nu$ and $\eta$.

Whereas (8) describes the motion of two clusters of sizes $N_1$ and $N_2 = N - N_1$, respectively, it says nothing about intra-cluster stability and cannot model the breakup of either cluster. To be able to evaluate the internal stability of the clusters, we followed Ku et al.[22] and added two effectively infinitesimal extra oscillators to the model, which only feel the presence of the two macroscopic clusters, but themselves neither affect the movement of each other, nor that of the macroscopic clusters. Their motion is given by

$$\begin{aligned}
\frac{dp_{1,2}}{dt} &= p_{1,2} - \nu q_{1,2} - \eta^2 \left[P_{1,2} - c_2 Q_{1,2}\right] \\
&\quad + \eta^2 \left[\frac{N_1}{N}(A_{c1} - c_2 B_{c1})\right. \\
&\quad \left. + \frac{N - N_1}{N}(A_{c2} - c_2 B_{c2})\right], \\
\frac{dp_{1,2}}{dt} &= q_{1,2} + \nu p_{1,2} - \eta^2 \left[Q_{1,2} + c_2 P_{1,2}\right] \\
&\quad + \eta^2 \left[\frac{N_1}{N}(B_{c1} + c_2 A_{c1})\right. \\
&\quad \left. + \frac{N - N_1}{N}(B_{c2} + c_2 A_{c2})\right],
\end{aligned} \tag{9}$$

where $P_{1,2}$ and $Q_{1,2}$ denote composite expressions for the first and second infinitesimal oscillator, of the same form as $A_{c1,c2}$ and $B_{c1,c2}$:

$$\begin{aligned}
P_{1,2} &:= 3p_{1,2} + 3(p_{1,2})^2 + (p_{1,2})^3 \\
&\quad + p_{1,2}(q_{1,2})^2 + (q_{1,2})^2, \\
Q_{1,2} &:= q_{1,2} + 2p_{1,2}q_{1,2} + (p_{1,2})^2 q_{1,2} + (q_{1,2})^3.
\end{aligned} \tag{10}$$

In the initial state of the continuation, one of these two infinitesimal oscillators is set to follow the same periodic orbit as either of the two clusters. If any bifurcations are detected to make either infinitesimal oscillator leave the macroscopic cluster it started at, this means that cluster has become unstable.

Figure 3a was created by initializing the $N = 16$ ensemble in the $8 - 4 - 4$ configuration at $c_2 = -0.715$ and incrementing $c_2$ by $\Delta c_2 = 10^{-5}$ every $\Delta T = 4 \cdot 10^4$ time steps until $c_2 = -0.7095$ for $\nu = 0.1$ and $\eta = 0.63$. At the beginning of each $c_2$ step, random numbers $\leq 10^{-6}$ were added to the real and imaginary part of each oscillator to provoke the breakup of potential unstable clusters. Maxima of Re($w_k$) were plotted for the last 2000 time steps of simulation at each $c_2$ steps.

The schematic in Fig. 3b was drawn based on automatically detected cluster sizes at each $c_2$ step in the aforementioned $c_2$-incremented simulation. These cluster sizes were determined by calculating the pairwise cross-correlations of the trajectories of all oscillators over the last 2000 time steps at each $c_2$ step, respectively. If the cross-correlation differed from 1 by less than $\epsilon = 10^{-8}$, the two oscillators were deemed to belong to the same cluster. To calculate the cross-correlations and obtain the clusters, we used SciPy's built-in `scipy.cluster.hierarchy.linkage` function.

The schematic in Fig. 3c was determined based on an analogous $c_2$-incremented simulations for $N = 32$, $\nu = 0.1$ and $\eta = 0.63$, initialized in the $16 - 16$ configuration at $c_2 = -0.74$. Here, $\Delta c_2 = 2 \cdot 10^{-5}$ and $\Delta T = 10^4$. The simulation

was performed until $c_2 = -0.712$, producing the result in Supplementary Fig. 1a. Clusters were calculated as in the $N = 16$ case based on the last 800 time steps of simulation at each $c_2$ step, producing Supplementary Fig. 2b.

The schematic in Fig. 3d was determined based on two analogous $c_2$-incremented simulations for $N = 256$, $v = 0.1$ and $\eta = 0.63$. In the first of these, the ensemble was initialized in the $128 - 64 - 64$ configuration at $c_2 = -0.7145$, from where $c_2$ was incremented by $\Delta c_2 = 10^{-5}$ every $\Delta T = 2 \cdot 10^4$ time steps until $c_2 = -0.7117$, producing the result in Supplementary Fig. 2a. In a second $c_2$-incremented simulation for $N = 256$, the ensemble was initialized at $c_2 = -0.71172$ in the $128 - 64 - 33 - 16 - 15$ configuration found there in the prior $N = 256$ simulation with $\Delta c_2 = 10^{-5}$. From there on, $c_2$ was incremented by $\Delta c_2 = 2 \cdot 10^{-7}$ every $\Delta T = 2 \cdot 10^4$ time steps until $c_2 = -0.7116$, producing the result in Supplementary Fig. 3a, b. For either simulation, clusters were calculated based on the last 800 time steps of simulation at each $c_2$ step, producing Supplementary Figs. 2b and 3b, c, respectively.

Figure 5 b–d were created by simulating the $16 - 9 - (7 \times 1)$ solution in Fig. 5a for $T = 10^6$ time steps. Every $10^4$ time steps, the pairwise cross-correlation between all oscillators was calculated over an interval of 800 time steps, and if the cross-correlation of two oscillators was found to be greater than $1 - \varepsilon$ for $\varepsilon = 10^{-8}$ (Fig. 5b), $\varepsilon = 10^{-4}$ (Fig. 5c) or $\varepsilon = 10^{-2}$ (Fig. 5d), respectively, they were counted as being in the same cluster.

The solutions in Fig. 6 were obtained by initializing the $N = 20$ ensemble in a $15 - 5$ solution at $c_2 = -0.88$, $v = 0.1$ and $\eta = 0.67$, and incrementing $c_2$ by $\Delta c_2 = 2 \cdot 10^{-4}$ every $\Delta T = 5000$ time steps until $c_2 = -0.7$. Supplementary Figures 4–7 were created based on data obtained analogously to that in Figs. 3 and 6 with parameters as given in their respective captions.

**Pitchfork maps**. The differential equations (1) were solved numerically using the Python programming language[46] (version 3.8). The data were held and processed in the form of NumPy (version 1.19) arrays[48] with complex-valued floating-point elements and visualized using the Matplotlib library and graphics environment (version 3.3) [49]. The numerical results were evaluated using custom-built functions drawing on the resources of these standard Python libraries, written by S.W.H.

The $\alpha$-incremented simulations behind Fig. 8 were initialized in the $N/2 - N/2$ configuration and $\alpha$ was then gradually increased as specified in the captions of Supplementary Figs. 9 and 10. The same applies to Supplementary Fig. 11. At the beginning of each $\alpha$ step, small random numbers $\leq 10^{-6}$ were added to the maps to provoke the breakup of potential unstable clusters.

The cluster sizes at each $\alpha$ step of the aforementioned simulations were calculated automatically by comparing the trajectories of all maps during the last 2000 steps at each $\alpha$ value. If the Euclidean distance between the vectors made up by two such map trajectories was found to be less than $\varepsilon = 10^{-4}$, the two maps were said to be in the same cluster.

**Electrochemical experiments**. For the experiments a custom made three electrode electrochemical PTFE cell is used, with a circular shaped platinum wire as counter electrode and a commercial mercury-mercurous sulfate reference electrode[34]. As working electrode a sample from an n-type Silicon wafer with a (111) crystal orientation and a resistivity of 1–10 Ω cm is used. A 200 nm aluminium back contact is evaporated onto the wafer and annealed at 250 °C for 30 min. To passivate the silicon surface and get rid of organic contamination, the samples are plasma-oxidized.

Before the experiment the sample is brought into contact with the wire in the custom made PTFE WE holder using silver paste. It is subsequently sealed using silicone rubber (Scrintex 901, Ralicks GmbH, Rees- Haldern, Germany), leaving free only the active electrode area. After the silicone has dried, the sample holder with the sample is immersed in acetone for 5 min, subsequently in ethanol for 5 min, then methanol for 5 min, then in ultra pure water ($R = 18.2$ MΩ cm) for 10 min and finally it is abundantly rinsed with ultra pure water.

The organic cleaning solvents are AnalaR NORMAPUR grade (VWR Chemicals). The electrolyte components are Suprapur grade (Merck). For the potential control a FHI-2740 potentiostat is used. Illumination of the electrode is provided by a 15 mW HeNe laser with a wavelength of 632.8 nm (Thorlabs HNL150L). The illumination intensity is controlled by an SLM (Hamamatsu x10468-06). The ellipsometric signal is recorded with a JAI-CV-A50 CCD camera. A background correction of the video data is performed according to

$$\xi(\vec{x}) = (\xi(\vec{x})_{raw} - \overline{\xi(\vec{x})}_{raw}) \cdot \frac{\langle \overline{\xi(\vec{x})}_{raw} \rangle}{\overline{\xi(\vec{x})}_{raw}}, \quad (11)$$

where $\xi(\vec{x})$ is the corrected ellipsometric signal at $\vec{x}$, $\xi(\vec{x})_{raw}$ is the raw data of the ellipsometric signal at $\vec{x}$, $\overline{\xi(\vec{x})}_{raw}$ is the temporal average of the raw data and $\langle \overline{\xi(\vec{x})}_{raw} \rangle$ denotes the spatial average of the temporal average of the raw data. The homogeneous mode is subtracted after the background correction is performed.

The experimental data presented are obtained from experiments under the following conditions: The electrolyte used for Fig. 9a, b had a pH of 2.3 and a fluoride concentration $c_F = 50$ mM. For (a) the applied potential was $U = 5.65$ V vs SHE, the external resistance times electrode area $R_{ext}A_{el} = 0$ kΩ cm$^2$ and the illumination intensity $I_{ill} = 0.67$ mW/cm$^2$. For (b) $U = 6.65$ V vs SHE, $R_{ext}A_{el} = 3.84$ kΩ cm$^2$, $I_{ill} = 0.57$ mW/cm$^2$. The electrolyte used for (c) and (d) had a pH of 1 and a fluoride concentration of $c_F = 75$ mM. For (c) the applied potential was $U = 8.65$ V vs SHE,

the external resistance times electrode area $R_{ext}A_{el} = 0.81$ kΩ cm$^2$ and the illumination intensity $I_{ill} =$ mW/cm$^2$. For (d) $U = 8.65$ V vs SHE, $R_{ext}A_{el} = 0.54$ kΩ cm$^2$, $I_{ill} = 0.57$ mW/cm$^2$. The illumination on the working electrode was homogeneous at any time.

## Data availability
The numerical and experimental data generated in this study have been deposited in the database of the TUM University Library under the accession code https://doi.org/10.14459/2021mp1618587.

## Code availability
The code is available as free and open source software under the GPL version 3 or later. It has been deposited in the database of the TUM University Library under the accession code https://doi.org/10.14459/2021mp1618587.

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

## Acknowledgements

We thank Felix P. Kemeth, Maximilian Patzauer and Seungjae Lee for fruitful discussions. Financial support from the Studienstiftung des deutschen Volkes and the Deutsche Forschungsgemeinschaft, project KR1189/18 "Chimera States and Beyond", is gratefully acknowledged.

## Author contributions

S.W.H. carried out the simulations and analysed the numerical data. A.T. performed the experiments and analysed the experimental data. All three authors discussed the results and wrote the paper. K.K. supervised the project.

## Funding

## Competing interests

The authors declare no competing interests.
