## [Peer Review File · Nature Communications]

Reviewers' Comments:

Reviewer #1:

Remarks to the Author:

I have found the main message of this paper to be extremely interesting and important. The Authors show how a system of Stuart-Landau oscillators with all-to-all coupling can bifurcate from order to disorder. It is extremely difficult to track bifurcations in such higher-order systems, at the same time we know how important are bifurcations in general to understand the behavior of dynamical systems. This is clearly a strength of the paper. Another point of force of the paper is that it focuses on Stuart-Landau oscillators, they represent a paradigmatic oscillator close to the Hopf bifurcation, such that the conclusions derived by the Authors have an importance not limited to a single specific example of dynamics. I therefore would like to support the publication of this paper.

The paper is well written, generally very clear and exhaustive. Still I think that there are a few important major points that I think could improve the clarity of the paper and/or provide further information to questions that, at least to me, seem to naturally originate from the discussed results and their presentation.

1) The Authors focused on the order \rightarrow disorder transition. However, one could ask what happens in the transition from disorder to order. In particular, all the bifurcations studied in the paper have been presented in this direction. However, it is very natural to ask what is the system behavior if we go in the other direction. Can we find at least some of the states reported in the paper also starting from a disordered scenario? What can be said of bifurcations occurring in the opposite direction with that studied in the paper? This seems to me also relevant (and indeed reading the first sentences of the introduction I was, somehow, expecting to find something also on the disorder to order transition).

2) I have found hard to follow the Section "HIERARCHICAL CLUSTERING THROUGH PERVASIVE STEPWISE SYMMETRY BREAKING". As a premise, I first want to clarify that this could be a personal feeling, but still I would like to invite the Authors to read again this section in light of my comments. On the contrary, there are sections in this paper which are extremely clear (and stimulating). Let me now go to the specific comments on this section. I had the feeling that trying to follow the many details that are in the section one may miss the overall picture. What I have found in general hard to follow is the interplay by the parameters η and c_2 . While in the first explanation on the model the focus was on the behavior vs η , then the discussion somewhat jumps to discuss to changes in c_2 . It was much easier to me to follow the considerations in terms of a decreasing value of the coupling, rather than on a parameter of the dynamics of the unit. I don't know if presenting a bifurcation diagram vs. η from the beginning could help the discussion. I would definitely suggest to add the time evolution of the variables corresponding to the cases illustrated in Fig. 1.

3) It is clear to me that the Authors can generate some states in the system of Stuart-Landau oscillators by exploring the parameter space following given directions (of parameter variations) that their analysis reveal. But what about the emergence of these states for fixed values of the parameters? Is there coexistence with other attractors? What can be said in terms of the basin of attractions of these states? This point is clearly also related to my point 1), as one can go through a bifurcation diagram in diverse direction, from order to disorder as the Authors did, or from disorder to order.

4) What can be said about natural/real systems showing behaviors similar to those here reported? Do the Authors can provide physical examples where their analysis is useful to explain the behavior observed?

5) I think it would be useful to add a DISCUSSION section, in which a comparison with other papers can also be dealt with. In particular, it seems to me a key of this work the nonlinear coupling used.

6) I have found also extremely interesting the discussion in the section EMERGENCE OF A

CHIMERA STATE. I agree that the analysis done here can be important to understand chimera states, but is this enough? Certainly, it is a big step forward, but understanding that these states can be generated from period doubling bifurcations and cluster splitting can also explain how these states originate in time, again for a fixed set of parameters? I think that the emergence of the chimera state from a set of initial conditions is a challenging question. Is there some link about the underlying bifurcation mechanism leading to this state and the way in which these states generate as the result of the evolution of the system starting from given initial conditions?

Reviewer #2:

Remarks to the Author:

The authors provide a detailed numerical investigation of the collective dynamical behavior displayed a network of Stuart-Landau oscillators with nonlinear, global coupling. The authors have discovered a very interesting bifurcation sequence from a state of global synchrony to a chimera state (with one large synchronized cluster of oscillators and many single incoherent oscillators) that passes through a series of cluster synchronized states. The authors perform a thorough analysis of each bifurcation in this sequence: first a series of symmetry breaking bifurcations, then a series of symmetry increasing bifurcations. The main bifurcation parameter is the coupling strength. The authors suggest that this may be a general bifurcation sequence for the creation of chimera states in globally coupled systems.

Over the last 20 years, there has been significant interest and research in chimera and cluster states. Interest in these areas continues to be high. Despite this, little progress has been made towards understanding the bifurcations that lead to chimera states. Some work has been done on bifurcations that lead to chimera states in structured networks (e.g., Panaggio, Abrams, Ashwin, and Laing. *Phys. Rev. E* 93, 012218), but this is the first work I am aware of that has analyzed bifurcations leading to chimera states in globally coupled systems.

The work seems to be technically correct, and the methods section is clear on how to reproduce the results, which are all numerical.

This work presents a new and interesting bifurcation sequence that is present in a network of Stuart-Landau oscillators with nonlinear global coupling. Detailed bifurcation sequences may seem rather technical and narrow in scope. However, this is a significant advance in understanding the relationship between cluster synchronization and chimera states and how they arise in networks. Further, there is at least some evidence (e.g., the authors refer to previous work on coupled maps) that this bifurcation sequence may be a more general phenomenon (i.e., it may occur in other systems). Therefore, I recommend the publication of this work in *Nature Communications* once the authors address the comments below.

The following few comments/questions may help the authors convey the generality of their result:

1. How strongly does this bifurcation sequence depend on the form of the nonlinear coupling? Why did the authors choose this form of nonlinear coupling? Nothing is mentioned about this in the manuscript.

2. It appears the authors have considered only even number of oscillators N . Is the bifurcation structure similar for odd N , even though the network cannot separate into, e.g., $N/2-N/2$? I suspect it would be similar, given the results in Fig. 3c, but the authors should confirm this. It may be good to include a comment about this for the interested reader.

3. The breaking up of clusters observed here seems to be closely related to isolated desynchronization (see Pecora, et al. *Nature Communications* 5.1 (2014): 1-8). Can the authors comment on if this is the case? If it is, a discussion of this relationship in the manuscript might provide additional context for the discussion of group and subgroup structure that appears in the conclusion section. Further, this may strengthen the argument that what the authors have discovered may be a general phenomenon, as the methods described in the Pecora paper have

been used for chimera states in Hart et al. Chaos 26. 9 (2016), 094801 and Cho et al. Phys. Rev. Lett, 119.8 (2017): 084101, although no bifurcation analyses were done in any of these reference.

4. In their description of Fig. 1e, the authors state that "For very unbalanced solutions, the smallest cluster is explosively destroyed in a subcritical pitchfork bifurcation." Does this result in a chimera state? Can the authors relate this chimera state with the one that is at the end of their hierarchical bifurcation sequence? Is it qualitatively the same type of chimera state?

5. "The larger N is, the more steps occur, at ever closer parameter values, and for N = 256, as many as seven steps can be observed (see Fig. 3 d)." ...is there any universality constant that describes the location of each bifurcation in parameter space?

6. (Optional) Figure 2c is a key figure, but I find it difficult to read. It would be nice if there were a way to label the stable state(s) in each region.

Reviewer #3:

Remarks to the Author:

This manuscript reports the results of detailed simulations and numerical studies of the bifurcations of a system of globally Stuart-Landau oscillators with nonlinear coupling as this coefficient is varied. A sequence of cluster-splitting period doubling bifurcations is followed by a torus bifurcation and then a symmetry increasing bifurcation.

The authors' main goal appears to be to demonstrate that a system of globally coupled oscillators displays "a range of coexistence patterns more comprehensive than chimeras."

After attempting to read the manuscript and assess the significance of the conclusions drawn it appears that the investigation is

(a) very technical in nature and will be appreciated by a very small segment of readers of Nature Communications;

(b) the system explored is highly specialized in terms of the nonlinear coupling term used. Previously this form of nonlinear coupling has been associated with certain electrochemical systems. There is no clear attempt in this manuscript to present the relevance of this model system to any experiments, past or present. The entire variation of c_2 is less than one percent in Fig 3!

(c) It is not apparent that there is universality in the results of this anecdotal numerical study. The sentence "This suggests that the bifurcation scenario uncovered here is universal to SN equivariant ensembles and possibly to other highly symmetric systems," is indeed just a suggestive statement. The dependence on N and trends for high N (the authors mention some results for N = 256 earlier in the manuscript) values will be interesting to explore.

Overall, it is strongly recommended that the authors submit the manuscript to a more specialized journal on dynamical systems or mathematics of nonlinear systems.

REVIEWER COMMENTS

Reviewer #1 (Remarks to the Author):

I have found the main message of this paper to be extremely interesting and important. The Authors show how a system of Stuart-Landau oscillators with all-to-all coupling can bifurcate from order to disorder. It is extremely difficult to track bifurcations in such higher-order systems, at the same time we know how important are bifurcations in general to understand the behavior of dynamical systems. This is clearly a strength of the paper. Another point of force of the paper is that it focuses on Stuart-Landau oscillators, they represent a paradigmatic oscillator close to the Hopf bifurcation, such that the conclusions derived by the Authors have an importance not limited to a single specific example of dynamics. I therefore would like to support the publication of this paper.

The paper is well written, generally very clear and exhaustive. Still I think that there are a few important major points that I think could improve the clarity of the paper and/or provide further information to questions that, at least to me, seem to naturally originate from the discussed results and their presentation.

1) The Authors focused on the order -> disorder transition. However, one could ask what happens in the transition from disorder to order. In particular, all the bifurcations studied in the paper have been presented in this direction. However, it is very natural to ask what is the system behavior if we go in the other direction. Can we find at least some of the states reported in the paper also starting from a disordered scenario? What can be said of bifurcations occurring in the opposite direction with that studied in the paper? This seems to me also relevant (and indeed reading the first sentences of the introduction I was, somehow, expecting to find something also on the disorder to order transition).

Reply: We thank the referee for this important question. In general, there is no significant hysteresis as long as the system only passes through supercritical bifurcations. This is e.g. the case for the $N=16$ c_2 -incremented simulation depicted in Fig. 3a. Inspired by the question, we have added an $N=16$ c_2 -decremented simulation backward along the same c_2 interval as a new Supplementary Fig. 4, referencing it accordingly in the main manuscript:

If we initialize the ensemble in the itinerant state beyond a symmetry-increasing bifurcation and gradually change the parameters back towards more regular motion, the transition to the relevant two-cluster solution will simply be the reverse of the one that created the itinerant state. For example, if we initialized the $N = 16$ ensemble in the state at the right edge of Fig. 3 a and slowly decreased c_2 , this would produce the same sequence of bifurcations. See Supplementary Fig. 4.

In some other cases, the currently observed solution branch can lose stability entirely, and the system will then jump to a different branch, hitherto co-stable with the originally studied one. We now state this in the paper after the above paragraph using the following words:

When the equation parameters are incremented too far into the regime of chaotic itinerancy, the ensemble will often jump to an entirely different solution. Beyond the 16-9-7-derived state in Fig. 5, it e.g. jumps to the blue hitherto co-stable 16-8-8-derived branch.

After such a jump from one solution branch to another, changing the parameter values back towards synchrony would indeed let us observe different bifurcations than on the originally studied branch. However, these are also just the forward (order-to-disorder) bifurcations of a different branch. For

example, the system might, as in the above quote, just have jumped from the branch of solutions emerging from the 16-9-7 solution to that of solutions emerging from the 16-8-8 solution. If we then decrease c_2 again from there on, we will also just observe the same backward succession of symmetry-increasing bifurcations and cluster-splitting cascade on the 16-8-8-derived branch. To our knowledge, there are no bifurcations that can fundamentally only be observed when going backward from disorder to order.

2) I have found hard to follow the Section "HIERARCHICAL CLUSTERING THROUGH PERVASIVE STEPWISE SYMMETRY BREAKING". As a premise, I first want to clarify that this could be a personal feeling, but still I would like to invite the Authors to read again this section in light of my comments. On the contrary, there are sections in this paper which are extremely clear (and stimulating). Let me now go to the specific comments on this section. I had the feeling that trying to follow the many details that are in the section one may miss the overall picture. What I have found in general hard to follow is the interplay by the parameters η and c_2 . While in the first explanation on the model the focus was on the behavior vs η , then the discussion somewhat jumps to discuss to changes in c_2 . It was much easier to me to follow the considerations in terms of a decreasing value of the coupling, rather than on a parameter of the dynamics of the unit. I don't know if presenting a bifurcation diagram vs. η from the beginning could help the discussion. I would definitely suggest to add the time evolution of the variables corresponding to the cases illustrated in Fig. 1.

Reply: We thank the referee for the comment and suggestion. The requested time-series plots have now been added to all subfigures in Fig. 1. Moreover, we have also added a full depiction (complex-plane plot and time series) of the 3-1 solution as it appears in the original frame of reference before eliminating the average oscillation. Together with the original-frame depiction of the 2-2 solution and the rotating-frame depictions of both these solutions, we hope it will better illustrate what the change to the co-rotating frame of reference accomplishes.

As for the interplay of the two parameters c_2 and η , these two parameters span a parameter plane, and the bifurcations discussed constitute curves in this plane, as shown in Fig. 2c. Thus, it is in general possible to cross any given bifurcation curve either by changing c_2 or changing η . Due to the orientation of the individual bifurcation curves, it is, however, sometimes more practical to change one parameter than the other in order to demonstrate a certain bifurcation sequence. We have now tried to make this relation between the parameters and the dependence of the bifurcations on both of them somewhat clearer, e.g. by changing the following sentence at the point in the text where we first mention the period-doubling bifurcation of the $N/2-N/2$ and emergence of the $N/2-N/4-N/4$ solution:

If we start at a point in parameter space where the $N/2-N/2$ solution is stable, keep $\eta < \eta_H$ sufficiently close to the Hopf bifurcation and gradually increase c_2 , one of the two clusters will break up into two smaller clusters.

This sentence now reads:

If we initialize the $N/2-N/2$ solution at a point in the $c_2 - \eta$ parameter plane where it is stable and from there on gradually change c_2 and/or η appropriately, one of the two clusters will break up into two smaller clusters.

In the same vein, we have added the following two introductory sentences to the beginning of the section "Hierarchical clustering...", when Fig. 2c is first mentioned in the text, about tracking this bifurcation in both c_2 and η :

If we concentrate on the $N/2-N/2$ solution, that is, keep $N_1/N = 0.5$ fixed, we can track the cluster-splitting period-doubling bifurcation in both c_2 and η simultaneously. A part of the resultant bifurcation line in the $c_2 - \eta$ parameter plane is delineated by the leftmost line in Fig. 2c.

Regarding the feeling that the overall picture might get lost in the many details of this section, we have now expanded and rewritten several sentences throughout the section for (hopefully) better understanding. We have also removed one and a half sentences on the 4-4-4-2-2 solution, which might be considered an expendable detail. (See the paragraph on page 4 beginning with “At the solid blue line...” Finally, we have indicated in Fig. 2c which of the discussed stable solutions are stable in which parts of the bifurcation diagram, hoping that this will make it easier to follow the description of these and the (partial) multistability between some of them.

3) It is clear to me that the Authors can generate some states in the system of Stuart-Landau oscillators by exploring the parameter space following given directions (of parameter variations) that their analysis reveal. But what about the emergence of these states for fixed values of the parameters? Is there coexistence with other attractors? What can be said in terms of the basin of attractions of these states? This point is clearly also related to my point 1), as one can go through a bifurcation diagram in diverse direction, from order to disorder as the Authors did, or from disorder to order.

Reply: In general, several different solutions coexist wherever the fully synchronized solution is unstable. In a large part of parameter space below the equivariant Hopf bifurcation of the synchronized solution at $\eta_H = 1/2^{0.5}$, several of these solutions are attracting, thus creating extensive multistability. We have now attempted to make this clearer by changing the beginning of the sentence at the start of the results section dealing with the Hopf bifurcation in which the synchronized solution is destabilized. Formerly, this read as follows:

The Hopf bifurcation occurs at $\eta_H = 1/2^{0.5}$ for suitable values of c_2 and v . For $v = 0.1$, which we keep fixed throughout, it does for $c_2 < -0.44817$. At η_H , several differently balanced two-cluster solutions emerge from the synchronized solution, some as stable and some as unstable, depending on the value of c_2 .

It has now been changed to the following, with a more explicit emphasis on the different cluster-size distributions of the two-cluster solutions emerging:

The Hopf bifurcation occurs at $\eta_H = 1/2^{0.5}$ for suitable values of c_2 and v . For $v = 0.1$, which we keep fixed throughout, it does for $c_2 < -0.44817$. In this Hopf bifurcation, differently balanced two-cluster solutions ranging from $(N-1) - 1$ (with all but one oscillator in the largest cluster) to $N/2-N/2$ (with half the oscillators in each cluster) emerge from the synchronized solution. Some of these emerge as stable and others as unstable, depending on the value of c_2 .

A little further in the text, we now also explicitly mention the $3N/4-N/4$ solution as a different example of such a two-cluster solution in addition to the $N/2-N/2$ solution:

An unbalanced $3N/4-N/4$ solution, with $N_1 = 3N/4$ of the oscillators in one cluster and $N_2 = N/4$ in the other, looks as in Fig. 1c-d.

We also hope that our new additional in-figure labeling of the stable solutions in Fig. 2c might contribute to a clearer understanding of how multistability is present in the described results.

As for the sizes and shapes of the basins of attraction of the different multistable solutions, trying to describe these goes beyond the scope of our paper. The same goes for any detailed study of which initial conditions lead to which solutions for given values of the parameters.

The focus of the section in question (and to a large part of our manuscript as a whole), is on the cluster-splitting cascade as a general mechanism – on how ever more complex solutions emerge in a stepwise process of successive discrete cluster splittings. Such a cascade exists for many (possibly all) of the different coexisting 2-cluster solutions. In an attempt to describe this mechanism more understandably, we initially focus on the concrete cascades that emerge when starting from $N/2$ - $N/2$ solutions for various values of N . Yet, Fig. 6 (formerly Fig. 5) shows an analogous cascade for a different initial cluster-size ratio of the two-cluster state, namely that of the $3N/4$ - $N/4$ solution. In order to further underline our focus on the general nature of the cascade, we have now added results on a different theoretical system, consisting of time-discrete maps and not of time-continuous oscillators. Finally, we now explicitly summarize the cluster-splitting cascade as a general phenomenon at the beginning of a new extended discussion section:

In this article, we have shown how a globally coupled system can transition from full symmetry to ever more complex coexistence patterns through a sequence of discrete symmetry-breaking steps. The transition begins with a cascade of cluster-splitting bifurcations, and at each step of this cascade, either one cluster or two similarly behaving clusters are split into smaller clusters. In an ideal form of the cascade, the next cluster to split is always the smallest one, ultimately creating a multi-cluster state with very different cluster sizes, wherein the smallest “cluster” is just a single oscillator. This ideal cascade is schematically depicted in the left part of Fig. 10.

4) What can be said about natural/real systems showing behaviors similar to those here reported? Do the Authors can provide physical examples where their analysis is useful to explain the behavior observed?

Reply: To our knowledge, an unquestionable cluster-splitting cascade and subsequent transition to symmetric turbulence via symmetry-increasing bifurcations, as discussed in our paper, has not been observed in any natural system yet. However, the photoelectrodissolution of n-type silicon does exhibit certain similarities. In these experiments, the development of the oxide layer on the electrode is studied using spatially resolved ellipsometry. The similarities to our scenario include the transition from a fully homogeneous electrode to one divided into modulated-amplitude anti-phase clusters, followed by another transition to partial subclustering – similar the part of our scenario wherein the oscillators transition from synchrony via the $N/2$ - $N/2$ to the $N/2$ - $N/4$ - $N/4$ solution. Moreover, the experimental system also exhibits a chimera state as well as fully turbulent oscillations throughout the electrode.

We have now included these experimental results, measured in our lab, at the end of our results section. Further details are found there as well as in the methods section. With the addition of these results, our paper has also acquired an additional co-author.

5) I think it would be useful to add a DISCUSSION section, in which a comparison with other papers can also be dealt with. In particular, it seems to me a key of this work the nonlinear coupling used.

Reply: We thank the referee for this suggestion and have now added an extensive discussion section, replacing the former “concluding remarks”. In this section, we discuss the common thread of our now three model systems and reflect upon the possible prerequisites for both the occurrence of stepwise symmetry breaking and symmetry-increasing bifurcations. In order to better depict and summarize our overall transition from synchrony to turbulence as a general phenomenon, we have also added a schematic of its different steps (Fig. 10). Finally, we draw connections to some previously studied numerical systems that might also exhibit our transition scenario.

6) *I have found also extremely interesting the discussion in the section EMERGENCE OF A CHIMERA STATE. I agree that the analysis done here can be important to understand chimera states, but is this enough? Certainly, it is a big step forward, but understanding that these states can be generated from period doubling bifurcations and cluster splitting can also explain how these states originate in time, again for a fixed set of parameters? I think that the emergence of the chimera state from a set of initial conditions is a challenging question. Is there some link about the underlying bifurcation mechanism leading to this state and the way in which these states generate as the result of the evolution of the system starting from given initial conditions?*

Reply: We also find the topic of the emergence of chimera states to be very interesting. This includes the question posed by the referee regarding the emergence of chimera states from different initial conditions for fixed parameter values. However, because of the intricate bifurcation structure (as explored in our manuscript) and high multistability of solutions in most of the part of parameter space where the fully synchronized solution is unstable, we must restrict ourselves to how chimeras emerge bifurcation-theoretically within the studied novel transition from order to disorder.

What can further be said from the given results, is e.g. that the 15-(5x1) chimera at $c_2 = -0.705$ and $\eta = 0.67$ in Fig. 6 is clearly multistable with the 10-10 solution. The latter is namely a type of $N/2$ - $N/2$ solution, and all $N/2$ - $N/2$ solutions are stable for these values of c_2 and η , as shown in the bifurcation diagram in Fig. 1k. Thus, initializing an $N = 20$ ensemble from random initial conditions here will at least sometimes lead to the 15-(5x1) chimera and sometimes to the 10-10 solution. In other cases, yet another solution will emerge, as will be shown in the following.

For the information of the referee, we include here four different sample simulations for $N = 20$ and different initial conditions – all of them conducted at the same parameter values $\nu = 0.1$, $c_2 = -0.705$ and $\eta = 0.67$, that is, the parameter values of the chimera state in Fig. 6e:

- Initializing the system close to a 10-10 configuration with weak superposed noise results in a transition to the 10-10 simply periodic (in the rotating frame) two-cluster solution:

- Initializing the system close to a 15-5 configuration with weak superposed noise results a transition to the 15-5 chimera state:

- Initializing the oscillators in a random cloud in one case results in a transition to a 12-(8x1) chimera state:

- A different random cloud of the same kind results in a transition to an 11-9 unbalanced simply periodic two-cluster solution:

What we infer from these preliminary simulations is a confirmation of the general insight that the system is strongly characterized by multistability. Overall, trying to chart the basin boundaries of the many attractors in the 40-dimensional phase space of only $N = 20$ oscillators thus seems to us to extend far beyond the scope of the present paper.

Reviewer #2 (Remarks to the Author):

The authors provide a detailed numerical investigation of the collective dynamical behavior displayed a network of Stuart-Landau oscillators with nonlinear, global coupling. The authors have discovered a very interesting bifurcation sequence from a state of global synchrony to a chimera state (with one large synchronized cluster of oscillators and many single incoherent oscillators) that passes through a series of cluster synchronized states. The authors perform a thorough analysis of each bifurcation in this sequence: first a series of symmetry breaking bifurcations, then a series of symmetry increasing bifurcations. The main bifurcation parameter is the coupling strength. The authors suggest that this may be a general bifurcation sequence for the creation of chimera states in globally coupled systems.

Over the last 20 years, there has been significant interest and research in chimera and cluster states. Interest in these areas continues to be high. Despite this, little progress has been made towards understanding the bifurcations that lead to chimera states. Some work has been done on bifurcations that lead to chimera states in structured networks (e.g., Panaggio, Abrams, Ashwin, and Laing. Phys. Rev. E 93, 012218), but this is the first work I am aware of that has analyzed bifurcations leading to chimera states in globally coupled systems.

The work seems to be technically correct, and the methods section is clear on how to reproduce the results, which are all numerical.

This work presents a new and interesting bifurcation sequence that is present in a network of Stuart-Landau oscillators with nonlinear global coupling. Detailed bifurcation sequences may seem rather technical and narrow in scope. However, this is a significant advance in understanding the relationship between cluster synchronization and chimera states and how they arise in networks. Further, there is at least some evidence (e.g., the authors refer to previous work on coupled maps) that this bifurcation sequence may be a more general phenomenon (i.e., it may occur in other systems). Therefore, I recommend the publication of this work in Nature Communications once the authors address the comments below.

Reply: We thank the referee for their favorable review. Furthermore, we are pleased to have been able to add additional results from different systems, something we believe further strengthens the case that the observed transition is rather general.

The following few comments/questions may help the authors convey the generality of their result:
1. How strongly does this bifurcation sequence depend on the form of the nonlinear coupling? Why did the authors choose this form of nonlinear coupling? Nothing is mentioned about this in the manuscript.

Reply: We have now added a few words on the origins of our main theoretical model to the introduction of the manuscript as follows:

Originally, this coupling was inspired by electrochemical experiments, wherein the oxide layer on a silicon electrode displays a wide range of spatiotemporal patterns. A few experimental measurements reminiscent of new results in equation (1) will be discussed later in this article.

As briefly referred to there, the manuscript now also contains a few results from the experimental system itself. The theoretical model was originally also spatially two-dimensional, developed on the

basis of the complex Ginzburg-Landau equation because of that equation's general nature and adapted to certain properties of the experiment. Later, it was reduced to an ensemble of purely globally coupled oscillators for reasons now mentioned in the beginning of the new subsection on our experimental results:

As stated in the introduction, equation (1) is inspired by electrochemical experiments. In fact, the theoretical model was originally more complicated, consisting not of discrete identical oscillators, but of a continuous oscillatory medium coupled via both global and local (diffusional) coupling. Later results showed that most of the qualitative dynamics obtained in this extended model could still be reproduced if the diffusion was set to zero, thus paving the way for our purely globally coupled ensemble. Meanwhile, the experimental system itself has been found to exhibit a vast amount of dynamical phenomena.

The general nature of the Stuart-Landau oscillator as well as the large variety of to some extent unconnected results previously obtained with these oscillators and nonlinear global coupling are also the reason why this kind of ensemble was chosen as the main model system of the present study.

As for the question of how the observed bifurcation sequence depends on the form of the nonlinear coupling, we now address this in the new discussion section as one among several possible prerequisites for our stepwise transition scenario from synchrony to turbulence. Moreover, we have added results showing an analogous transition of the dynamics in globally coupled time-discrete one-dimensional maps.

2. It appears the authors have considered only even number of oscillators N . Is the bifurcation structure similar for odd N , even though the network cannot separate into, e.g., $N/2$ - $N/2$? I suspect it would be similar, given the results in Fig. 3c, but the authors should confirm this. It may be good to include a comment about this for the interested reader.

Reply: The suspicion of the referee is correct. For odd N one does indeed observe a similar bifurcation structure. We have included a c2-incremented simulation for $N=63$ in the supplementary information (Suppl. Fig. 13), including a cluster-splitting cascade and two symmetry-increasing bifurcations. This figure is referenced briefly in the discussion when discussing the generality of the results:

The cascade is also not dependent on the choice of $N = 2n$, but similarly occurs for odd N as well, as shown in Supplementary Figure 13.

3. The breaking up of clusters observed here seems to be closely related to isolated desynchronization (see Pecora, et al. Nature Communications 5.1 (2014): 1-8). Can the authors comment on if this is the case? If it is, a discussion of this relationship in the manuscript might provide additional context for the discussion of group and subgroup structure that appears in the conclusion section. Further, this may strengthen the argument that what the authors have discovered may be a general phenomenon, as the methods described in the Pecora paper have been used for chimera states in Hart et al. Chaos 26. 9 (2016), 094801 and Cho et al. Phys. Rev. Lett, 119.8 (2017): 084101, although no bifurcation analyses were done in any of these reference.

Reply: It seems very likely that at least some of the cluster splitting bifurcations are examples of 'isolated desynchronization', i.e., the breakup of certain clusters without disturbing the others.

However, to conduct an analysis as derived by Pecora et al. (or Cho et al.), where the stability of all the dynamically valid cluster synchronization patterns is determined, is a formidable task in our case. This is due to both the high symmetry of the network (which has full S_N permutation symmetry) and the large number of oscillators N in the network considered. With $N=6$ oscillators there are 202 possible cluster synchronization patterns of the complete graph, and already for $N=8$ oscillators, their number becomes excessive and it would be extremely challenging to handle the problem numerically.

The smallest ensemble we studied in detail has $N=16$ (see Fig. 2). For a comparison, in the mentioned chimera papers, networks with 4 (Hart et al.) and 6 (Cho et al.) oscillators were considered, respectively. We consider it one of the strengths of our manuscript to present a bifurcation analysis for large, globally coupled ensembles. Moreover, we believe we have further strengthened the argument that the cluster splitting cascade is a general phenomenon by presenting it in an additional, unrelated model as well as by detecting parts of it experimentally.

4. *In their description of Fig. 1e, the authors state that “For very unbalanced solutions, the smallest cluster is explosively destroyed in a subcritical pitchfork bifurcation.” Does this result in a chimera state? Can the authors relate this chimera state with the one that is at the end of their hierarchical bifurcation sequence? Is it qualitatively the same type of chimera state?*

Reply: Unfortunately, our choice of the wording “explosively destroyed” was misleading, as the outcome is not always only single oscillators, but depends on the relative sizes of the two clusters before crossing the subcritical pitchfork bifurcation. The important property of this bifurcation is actually that it is subcritical and that the resultant state after crossing the bifurcation line is not directly related to the initial two-cluster state, but rather belongs to a coexisting different branch. We have removed the word “explosively” and clarified how variable the outcome of the subcritical transition is. The paragraph of the results section dealing with this bifurcation now reads as follows:

For very unbalanced solutions $N_1/N > 0.8$, the smallest cluster is destroyed in a subcritical pitchfork bifurcation (green line). This results in several smaller clusters and/or single oscillators, depending on the relative sizes of the initial two clusters. In some cases, a few oscillators originally in the smaller cluster are also absorbed by the larger one. As the transition is subcritical, these outcome states are not directly related to the initial two-cluster solution, but rather belong to a different, coexisting solution branch. They will not concern us further here.

5. *“The larger N is, the more steps occur, at ever closer parameter values, and for $N = 256$, as many as seven steps can be observed (see Fig. 3 d).” ...is there any universality constant that describes the location of each bifurcation in parameter space?*

Reply: We could not find any such universality constant. In the system as we have studied it, such a constant cannot be identified because of the way most bifurcation curves are parametrized by more than one parameter. Thus, the distance traveled e.g. between any one period-doubling bifurcation in Fig. 2c and the next depends both on where one starts on the first period-doubling curve and on the direction in the c_2 - η plane in which one travels. Possibly, a hypothetical future renormalization of our system might yield a universality constant, or it might be possible to find such a constant in a simpler model than ours also displaying the cluster-splitting cascade.

6. (Optional) Figure 2c is a key figure, but I find it difficult to read. It would be nice if there were a way to label the stable state(s) in each region.

Reply: We find this a very helpful suggestion and have now labeled the stable states derived from the 8-8 solution in each region of the diagram in Fig. 2c.

Reviewer #3 (Remarks to the Author):

This manuscript reports the results of detailed simulations and numerical studies of the bifurcations of a system of globally Stuart-Landau oscillators with nonlinear coupling as this coefficient is varied. A sequence of cluster-splitting period doubling bifurcations is followed by a torus bifurcation and then a symmetry increasing bifurcation.

The authors' main goal appears to be to demonstrate that a system of globally coupled oscillators displays "a range of coexistence patterns more comprehensive than chimeras."

Reply: We fear this is a misunderstanding. Our focus is on the dynamical stepwise transition scenario from synchrony via more complex cluster states to turbulence. It is not on the system of Stuart-Landau oscillators as such, nor on the "mere" fact that the system exhibits a wealth of coexistence patterns. In the course of an extensive revision, we have added additional numerical and experimental data from different systems and hope that this will contribute to making the intended topic of our paper clearer. Moreover, we have changed the former short "concluding remarks" into a significantly longer discussion section wherein we discuss the possible prerequisites for the observed transition scenario.

After attempting to read the manuscript and assess the significance of the conclusions drawn it appears that the investigation is

(a) very technical in nature and will be appreciated by a very small segment of readers of Nature Communications;

Reply: Attempting to address these concerns, we have attentively gone through all reviewer comments as well as the text in its entirety and reformulated and expanded several paragraphs throughout, with the intention to improve readability.

In an attempt to further increase the understandability of the paper, we have added an additional schematic (Fig. 4) depicting the essence of what a symmetry-increasing bifurcation is, as well as a schematic of the overall transition from synchrony to turbulence and its different steps (Fig. 10).

(b) the system explored is highly specialized in terms of the nonlinear coupling term used. Previously this form of nonlinear coupling has been associated with certain electrochemical systems. There is no clear attempt in this manuscript to present the relevance of this model system to any experiments, past or present. The entire variation of c_2 is less than one percent in Fig 3!

Reply: As addressed in our reply to referee #2, we have now included more information on the link between the electrochemical experiments providing the background of our main model and that model itself. Moreover, we now also provide certain tentative results in the electrochemical experiments themselves, resembling parts of the described transition scenario.

As for the small variation of c_2 in Fig. 3, this is a consequence of the fact that the further one precedes in the studied cluster-splitting cascade, the closer do subsequent period-doubling cluster splits lie. The parameter interval from the second period-doubling bifurcation to the end of the cascade, as depicted e.g. in Fig. 3a, is thus almost bound to be rather short. If we had instead begun the depicted simulation interval at the first period-doubling of the 8-8 branch, that is, at the leftmost bifurcation curve in Fig. 2c, the overall parameter variation would have been significantly larger, and if we had begun depicting the transition at the Hopf bifurcation from full synchrony, the variation would have been larger still. That would, however, have been at a cost to the resolution towards the far end of the cascade (and around the symmetry-increasing bifurcation) depicted in Fig. 3a.

In Fig. 6 (formerly Fig. 5), the difference in c_2 from the first to the last step of the cascade depicted there, that is, from the depicted 15-5 solution to the depicted 15-(5x1) chimera is a full 0.17. Furthermore, this figure suggests that the cluster-splitting cascade is indeed prevalent in this model, as it occurs there for relative sizes of the two initial clusters that are very different from $N/2-N/2$. As addressed above, we have now also added similar results in time-discrete coupled maps, demonstrating that the observed transition is less specialized than the first version of our manuscript might have suggested.

(c) It is not apparent that there is universality in the results of this anecdotal numerical study. The sentence "This suggests that the bifurcation scenario uncovered here is universal to SN equivariant ensembles and possibly to other highly symmetric systems," is indeed just a suggestive statement. The dependence on N and trends for high N (the authors mention some results for $N = 256$ earlier in the manuscript) values will be interesting to explore.

Reply: We agree with the referee that the sentence they quote here is more suggestive than we intended it to. Within the context of a revised and significantly expanded discussion section, we have changed the sentence to refer more specifically to the globally coupled logistic maps of Kaneko. In its modified form, it is now found at the end of the following paragraph, dealing with both the points addressed in our reply to the referee's previous concern and the logistic maps:

Symmetry-increasing bifurcations also occur whether the cluster-size distribution of the initial two-cluster state emerging from synchrony is $N_1/N_2 = 1$ (Fig. 3 a) or $N_1/N_2 = 3$ (Fig. 6). Moreover, it is observed in both the Stuart-Landau oscillators (1) and the pitchfork maps (4). The general outcome of a symmetry-increasing bifurcation, chaotic itinerancy, has previously been found in globally coupled logistic maps, along with both multi-cluster states, chimeras and precision-dependent clustering, but without an overall explanation of how these phenomena might be bifurcation-theoretically related to each other. This suggests that the bifurcation scenario uncovered here occurs in those logistic maps as well.

Especially in the light of our own additional results obtained in coupled pitchfork maps, we believe at least the new version of this sentence is justified.

Overall, it is strongly recommended that the authors submit the manuscript to a more specialized journal on dynamical systems or mathematics of nonlinear systems.

Reply: We see how the first version of our manuscript might have led to such a conclusion and hope that the additional results and overall revision of the presentation and discussion has addressed this concern.

Reviewers' Comments:

Reviewer #1:

Remarks to the Author:

The Authors have revised their manuscript in a very detailed way. The answers that they have provided to my comments are convincing and improved the paper in several aspects. I asked to clarify a few paragraphs and address some curiosities and questions and this was done very accurately. In addition, I think that the new results presented add further value to an already interesting paper. The proof of generality of the results also appearing in the discrete-time domain and the experimental findings now included in the second version of the manuscript are valuable and fascinating. For these reasons, I would like to recommend the paper for publication as it is.

Reviewer #2:

Remarks to the Author:

In my opinion, the authors have improved the manuscript. I support its publication in Nature Communications.

Reviewer #3:

Remarks to the Author:

The authors have made a serious and successful effort to revise and enhance the accessibility of the manuscript to a larger group of readers. The manuscript is now much better organized and more informative, and should be published in Nat Comm

Reviewer #1 (Remarks to the Author):

The Authors have revised their manuscript in a very detailed way. The answers that they have provided to my comments are convincing and improved the paper in several aspects. I asked to clarify a few paragraphs and address some curiosities and questions and this was done very accurately. In addition, I think that the new results presented add further value to an already interesting paper. The proof of generality of the results also appearing in the discrete-time domain and the experimental findings now included in the second version of the manuscript are valuable and fascinating. For these reasons, I would like to recommend the paper for publication as it is.

Reviewer #2 (Remarks to the Author):

In my opinion, the authors have improved the manuscript. I support its publication in Nature Communications.

Reviewer #3 (Remarks to the Author):

The authors have made a serious and successful effort to revise and enhance the accessibility of the manuscript to a larger group of readers. The manuscript is now much better organized and more informative, and should be published in Nat Comm

Reply: *We thank all reviewers for their helpful comments as well as for their second review of our manuscript.*